# Order-of-magnitude enhancement in photocurrent generation of *Synechocystis* sp. PCC 6803 by outer membrane deprivation

Shoko Kusama [1,2], Seiji Kojima [2✉], Ken Kimura[1], Ginga Shimakawa[3], Chikahiro Miyake[4], Kenya Tanaka [1], Yasuaki Okumura[2] & Shuji Nakanishi [1,3,5✉]

Biophotovoltaics (BPV) generates electricity from reducing equivalent(s) produced by photosynthetic organisms by exploiting a phenomenon called extracellular electron transfer (EET), where reducing equivalent(s) is transferred to external electron acceptors. Although cyanobacteria have been extensively studied for BPV because of their high photosynthetic activity and ease of handling, their low EET activity poses a limitation. Here, we show an order-of-magnitude enhancement in photocurrent generation of the cyanobacterium *Synechocystis* sp. PCC 6803 by deprivation of the outer membrane, where electrons are suggested to stem from pathway(s) downstream of photosystem I. A marked enhancement of EET activity itself is verified by rapid reduction of exogenous electron acceptor, ferricyanide. The extracellular organic substances, including reducing equivalent(s), produced by this cyanobacterium serve as respiratory substrates for other heterotrophic bacteria. These findings demonstrate that the outer membrane is a barrier that limits EET. Therefore, depriving this membrane is an effective approach to exploit the cyanobacterial reducing equivalent(s).

[1] Graduate School of Engineering Science, Osaka University, 1-3 Machikaneyama, Toyonaka, Osaka 560-8631, Japan. [2] Panasonic Holdings Corporation, Kyoto 619-0237, Japan. [3] Research Center for Solar Energy Chemistry, Osaka University, 1-3 Machikaneyama, Toyonaka, Osaka 560-8631, Japan. [4] Graduate School of Agricultural Science, Kobe University, Nada-ku, Kobe 657-8501, Japan. [5] Innovative Catalysis Science Division, Institute for Open and Transdisciplinary Research Initiatives (ICS-OTRI), Osaka University, Suita, Osaka 565-0871, Japan. ✉email: kojima.seiji001@jp.panasonic.com; nakanishi.shuji.es@osaka-u.ac.jp

Recently, various technologies have been developed to utilize reducing equivalents produced by oxygenic photosynthesis, which absorbs light and oxidizes water to produce high energy electrons. Emerging among them is biophotovoltaics (BPV), in which the reducing equivalents are transferred extracellularly to electrodes or to other bacteria, a phenomenon called extracellular electron transfer (EET), eventually generating electrical power[1–4]. Cyanobacteria, which are gram-negative bacteria capable of performing oxygenic photosynthesis, have been extensively studied in the field of BPV research[1–3]. This is because cyanobacteria have higher energy conversion efficiency of photosynthesis than terrestrial plants, much like eukaryotic algae[5–7]. Moreover, they are easy to culture, amenable to genetic manipulation[8], grow fast, and possess a simple cell structure compared with eukaryotes. Many studies on the mechanism of EET have been conducted using cyanobacteria[9–15], and so far, direct EET via conductive nanowires[16] and indirect, mediated EET by endogenous mediators[14,15,17] have been put forward as possible EET pathways; some suggest that the latter is more likely than the former to occur in the case of cyanobacteria[2,18].

Although much progress has been made in the field of BPV in the past decade, the low EET activity of cyanobacteria remains a limitation. The EET activity of cyanobacteria, both in the dark and under illumination, is markedly lower compared with that of mineral-reducing, electricity-generating bacteria, e.g. the genera *Shewanella* and *Geobacter*[2], which are capable of utilizing diverse electron acceptors including an anode[19]. As previously pointed out, the main mechanism for the low EET activity of cyanobacteria is their autotrophy, in which EET could be totally useless and wasteful because electrons originating from phosynthesis should be fully utilized to provide enough reducing equivalents and energy to fix carbon[1]. Here, we hypothesize an additional mechanism for the low EET activity of cyanobacteria: the low permeability of the outer membrane. The outer membrane of cyanobacteria exhibits more than 20-fold lower permeability to organic substrates than that of *Escherichia coli*, the model gram-negative bacteria[20]. This low permeability is thought to reflect its autotrophic life style[20,21], which does not necessarily rely on uptake of extracellular nutrients, although various transport systems do exist and function in cyanobacteria[22–24].

Here, we show that, using an outer membrane-deprived *Synechocystis* sp. PCC 6803 (hereafter *Synechocystis*) mutant, slr0688i, in which the interaction between the outer membrane and the peptidoglycan is weakened so that the outer membrane is detached from the cell, a significant enhancement in cyanobacterial EET activity is achievable. EET activity is evaluated as extensively as possible in terms of photocurrent generation, ferricyanide reduction, and electron donation capacity as respiratory substrates (Fig. 1a). This study verifies our hypothesis that the low permeability of the outer membrane contributes to the low EET activity of cyanobacteria.

## Results
**Mediated EET is enhanced by outer membrane deprivation.** First, to compare the EET activity of the outer membrane-deprived mutant, slr0688i[25], with that of the wild type, dCas9, the photocurrent from both strains was measured. The growth curves of both cells and the typical electron-micrograph showing the outer membrane detachment of slr0688i cells are shown in Supplementary Fig. 1. Slr0688i and dCas9 resuspended in their respective supernatants were injected by gravity onto flat indium tin oxide (ITO) electrodes to which +0.25 V vs. Ag/AgCl was applied, and the current generated under illumination was recorded. Photocurrent generated from slr0688i cells was increased by an order of magnitude compared to dCas9 (Fig. 1b).

On average, slr0688i generated approx. 40 and 20 times as much photocurrent as dCas9 at the initial stage and steady state (at $t = 64$ and 100 s in Fig. 1b; Supplementary Table. 1), respectively, with flat ITO electrodes; respresentative amperometric i-t curves of both strains are shown in Fig. 1c. To confirm that the observed enhancement in photocurrent generation of slr0688i did not stem from increased photosynthetic activity, the oxygen evolving activities of slr0688i and dCas9 were measured and compared. Under illumination with high light (750 μmol photons $m^{-2}$ $s^{-1}$), neither photosynthetic nor respiratory activity was significantly different between them on a chlorophyll basis (Supplementary Fig. 2); moreover, under illumination with moderate light (120 μmol photons $m^{-2}$ $s^{-1}$), the photosynthetic activity of slr0688i was slightly lower than that of dCas9 (Supplementary Fig. 2). These results indicated that the enhanced photocurrent generation of slr0688i is not attributable to increased photosynthetic activity but to its outer membrane deprivation. As hydrophobicity of the cell surface is one of the decisive factors for the cell–electrode interaction, we performed a hydrophobicity assay[26] by mixing slr0688i or dCas9 cell cultures with *p*-xylene. It was revealed that $49.0\% \pm 8.7\%$ of dCas9 cells were adsorbed to *p*-xylene, whereas only $6.3\% \pm 1.8\%$ of slr0688i cells were adsorbed (Supplementary Fig. 3). The finding that the hydrophobicity of slr0688i is significantly lower than that of dCas9, and the fact that hydrophobic cells are more adhesive to the ITO substrate[27,28] suggest that cell–electrode interaction was not enhanced in the slr0688i mutant. Furthermore, we measured photocurrent under conditions where the influence of cell-electrode interaction was minimized by stirring the cell suspension in the presence of non-cell permeable mediator, ferricyanide (Supplementary Fig. 4). We confirmed that slr0688i cells generated significantly larger photocurrent than dCas9 cells. Therefore, the observed enhancement in current generation was attributable to outer membrane deprivation, not to enhanced cell–electrode interaction.

Next, to examine the necessity of supernatants in photocurrent generation, the supernatant of slr0688i was substituted with that of dCas9 or fresh BG11, and the photocurrent was measured. Photocurrent generation from slr0688i was not affected by substitution with the supernatant of dCas9 (Fig. 1d; red versus pink lines) but was abolished by substitution with fresh BG11 (Fig. 1d; red versus green lines). In contrast, dCas9 did not generate a photocurrent with the supernatant of either slr0688i ((Fig. 1d; blue line) or dCas9 (Fig. 1d; black line) under our experimental conditions. These results demonstrated that redox-active, secreted compounds included in the supernatants of slr0688i and dCas9 serve as electron mediators in photocurrent generation from slr0688i. Some anodic peaks were observable in cyclic voltammograms taken from slr0688i resuspended in its supernatant (Supplementary Fig. 5), which are possibly attributable to redox-active compounds secreted in the supernatant. These observations are consistent with recent suggestions that cyanobacterial secreted compounds serve as endogenous electron mediators in photocurrent generation from wild-type cells[2,14,15]. In addition, photocurrent from the cells was observed with the supernatant of slr0688i containing compounds of molecular weight (MW) < 3000 but not of MW > 3000 (Supplementary Fig. 6), which is consistent with the prediction that small molecular compounds, such as quinones and flavines, might be responsible for photocurrent generation[2]. Taken together, these results indicated that outer membrane deprivation facilitates mediated EET, which requires secreted redox-active compounds regardless of deprivation of the outer membrane.

As the level of photocurrent generation by BPV system is determined not only by EET activity of the cell but also by the properties of electrodes, a vast improvement of photocurrent generation is readily expected by utilizing slr0688i cells and

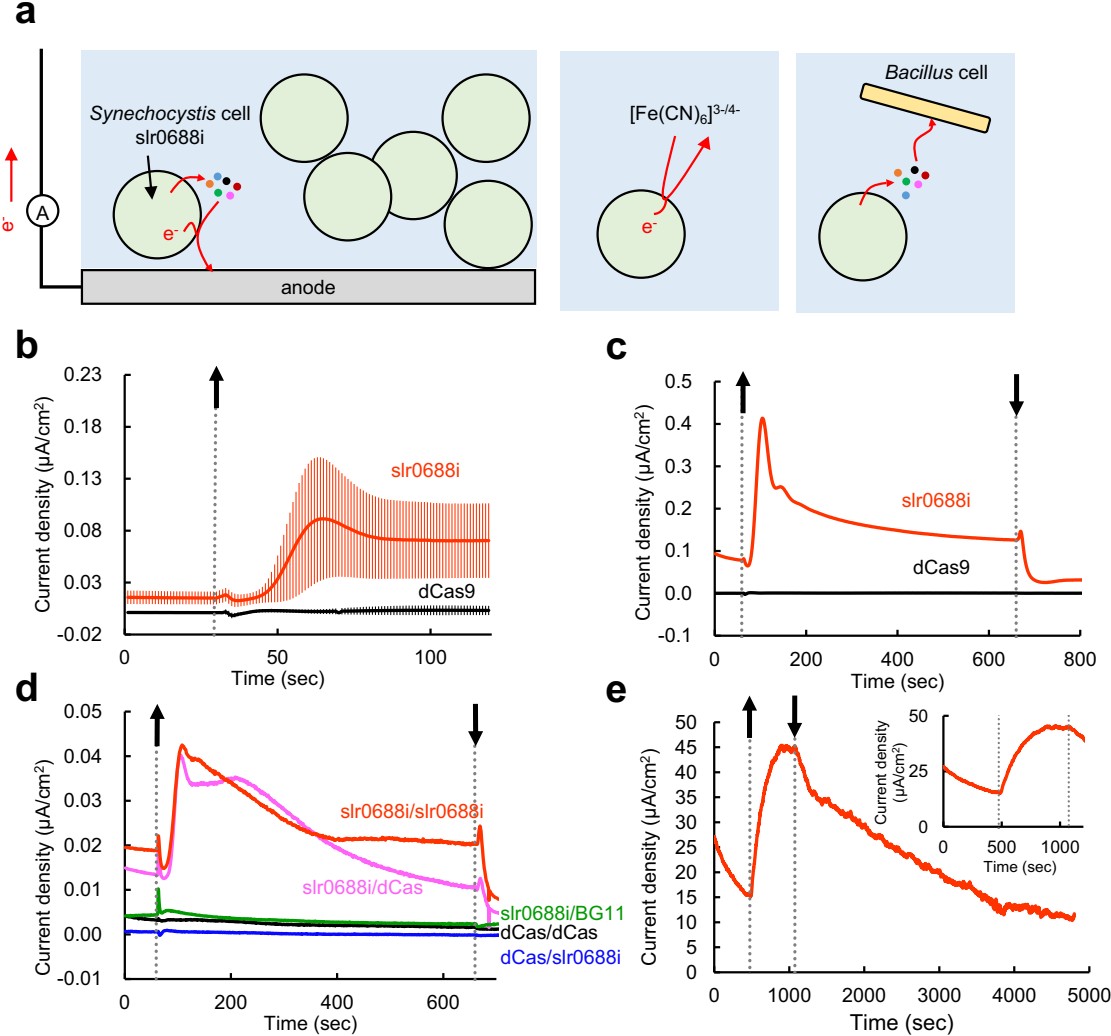

**Fig. 1 EET and photocurrent generation from *Synechocystis* cells. a** Schematic summary of EET from outer membrane-deprived *Synechocystis* cells, slr0688i. From slr0688i, reducing equivalents could be transferred to an exterior electrode via secreted compounds (mediated EET) or to an artificial electron acceptor, ferricyanide (ferricyanide-mediated EET). In addition, the extracellularly derived reducing equivalents could serve as respiratory substrates for other bacteria (here, *Bacillus cereus*), i.e., electron donors in BPV systems. **b** Slr0688i (OD$_{730}$ = 1.5, 4 mL; red points) and dCas9 (OD$_{730}$ = 1.5, 4 mL; black points) were injected by gravity onto plane ITO electrodes, and +0.25 V vs Ag/AgCl was applied to obtain chronoamperograms. Averages ± 2 SE from 33 and 21 biological replicates for slr0688i and dCas9, respectively, are presented. **c** Representative chronoamperograms of slr0688i (OD$_{730}$ = 1.5, 4 mL; red line) and dCas9 (OD$_{730}$ = 1.5, 4 mL; black line). **d** The supernatant of slr0688i was substituted either with that of dCas9 (pink line) or with fresh BG11 medium (green line). Also shown as controls are slr0688i and dCas9 suspended in their respective supernatants (red and black lines, respectively), and dCas9 in slr0688i supernatant (blue line). All cell suspensions were adjusted to OD$_{730}$ = 1.5 (4 mL) and injected onto flat ITO, and +0.25 V vs Ag/AgCl was applied. **e** Slr0688i (OD$_{730}$ = 14.6, 4 mL) after 6 days of culture were injected by gravity onto a piece of carbon paper placed upon flat ITO, and +0.25 V vs Ag/AgCl was applied. The inset shows the enlarged view from 0 to 1200 s of the measurement. The upward and downward arrowheads indicate the beginning and end of illumination (**b**–**d**, 100 μmol photons m$^{-2}$ s$^{-1}$; e, 420 μmol photons m$^{-2}$ s$^{-1}$), respectively. Source data are provided as a Source Data file.

electrode(s) more efficient than flat ITO. To verify this assumption, we utilized a carbon paper (CP) electrode as an anode, because (i) it provides high porosity and high specific area[29] which increase chances of electron donation from extracellular mediator(s) to the electrode and (ii) carbon-based electrodes are empirically known to generate relatively high photocurrent in cyanobacterial BPV[13,15,17,30,31]. Indeed, we observed approx. 30 μA/cm$^2$ photocurrent generation by slr0688i cells when using a CP anode (Fig. 1e; Supplementary Fig. 7). This was more than 100 times higher than that measured with a flat ITO anode, and represents the highest photocurrent by means of genetic manipulation of the cyanobacterial cell[31]. Also, this photocurrent amplitude is the highest among the previously

reported values of BPV systems using the untreated whole cells of *Synechocystis*[12–14,32–34]; note that Saper et al.[15] reported that a mild physical pre-treatment of *Synechocystis* cells enhances the electrogenic activity and recorded appx. 30 μA/cm$^2$. It should be noted that we employed the dense cell culture obtained after a prolonged cultivation period, i.e., 6-days cultivated cell culture, to achieve the high photocurrent value as above, but that we also noticed that this dense and 'aged' cell culture sometimes produced unstable photocurrent for unknown reason. As such, we used dilute and 'young' cell culture, i.e., 3-days cultivated cell culture, which tended to produce more stable photocurrent, for other comparative analysis with CP anodes such as the comparison of photocurrent between slr0688i and dCas9 cells

or investigation of the effects of photosynthesis inhibitors (Supplementary Fig. 8a and 15, respectively). We confirmed that dCas9 cells produced only a modest level of photocurrent compared to slr0688i cells even when using a CP anode (Supplementary Fig. 8a). This result assures that the high photocurrent level of slr0688i cells is attributed to the outer membrane deprivation but not to the property of the CP anode. It seemed noteworthy, however, that the photocurrent profile observed with a CP anode may not completely the same as that with an ITO anode, as we noticed that slr0688i cells generated photocurrent (approx. 2 to $3 \mu A/cm^2$) even after replacing the culture supernatant to fresh BG11 medium (Supplementary Fig. 8b), on the contrary to the fact that this treatment completely abolished the photocurrent when using an ITO anode (Fig. 1d; Supplementary Fig. 8b). Hence, the photocurrent generated by slr0688i cells with a CP anode might stem partially from EET pathway(s) distinct from that to an ITO anode, e.g., the one via direct interaction between the cells and the electrode. To focus solely on the effect of outer membrane deprivation on EET activity, ITO was used as an anode in all the following experiments.

**Electrons are suggested to stem from pathway(s) downstream of photosystem I**. To narrow down the site where the reducing equivalents were transferred extracellularly, we examined the effects of a series of photosynthesis inhibitors, namely 3-(3,4-dichlorophenyl)-1,1-dimethylurea (DCMU), potassium cyanide (KCN), phenylmercuric acetate (PMA), glycolaldehyde (GA), and $p$-chloromercuribenzoate (pCMB), on photocurrent generation by slr0688i cells. Each of these compounds inhibits different sites of photosynthetic reactions ranging from photosystem (PS) II to the downstream of PSI[35–46] (Fig. 2b). Surprisingly, only pCMB markedly enhanced current generation, whereas the other inhibitors eliminated or significantly decreased the current under illumination (Fig. 2a, Supplementary Fig. 9). The pCMB-enhanced photocurrent was most pronounced shortly after the start of illumination, and the photocurrent subsequently declined close to the initial background level within 10 min. We confirmed that this photocurrent pattern was reproducible (biological repeats of the measurements are shown in Supplementary Fig. 9e).

The action of the aforementioned inhibitors except DCMU, which inhibits electron transfer from plastoquinone (PQ), $Q_A$, to another PQ, $Q_B$, in PSII[40,41], is pleiotropic and often ambiguous in vivo. We therefore employed multiple photosynthetic parameters to specify the action of pCMB and to distinguish it from those of the other inhibitors (Supplementary Figs. 10–13). The pulse amplitude modulated (PAM) measurements and NADPH fluorescence revealed that pCMB did not affect the light-dependent redox kinetics of P700 (Supplementary Fig. 12a) nor inhibit NADPH generation upon illumination (Supplementary Fig. 13a). These results indicated that pCMB inhibited the reaction downstream of photosynthetic NADPH generation. While NADPH generation was not inhibited at the onset of light illumination, prolonged light illumination (>1 min) of pCMB-treated cells caused a decrease in oxygen-evolving activity and effective quantum yield of PSII [Y(II)] (Supplementary Figs. 10a, 11b). These phenomena are common in cells in which the Calvin cycle is inhibited[47]. These observations suggest that pCMB perturbs the consumption of photosynthetically generated reducing energy presumably by inhibiting the Calvin cycle, which may temporarily reinforce the reduction of possible electron mediator(s) as the alternative consumption route, thereby enhancing the photocurrent.

As it was necessary to distinguish the action of the other inhibitors from that of pCMB, we likewise evaluated their effects on photosynthetic parameters. DCMU unequivocally inhibited electron transfer from $Q_A$ to $Q_B$ in PSII[40,41], which was evident from the fact that Y(II) became zero from the onset of measurements (Supplementary Fig. 10b). KCN at 5 mM completely inhibited both oxygen consumption and evolution activity measured with $NaHCO_3$ as the sole electron acceptor (Supplementary Fig. 11a); this result was consistent with its reported inhibitory effects on respiratory terminal oxidases[43,44,46] as well as plastocyanin[35,37,38] and possibly the Calvin cycle[48]. PMA induced decreases in Y(II) and oxygen-evolving activity upon illumination in a similar manner to pCMB (Supplementary Figs. 10b, 11c), but NADPH generation and consumption upon illumination and in the dark, respectively, were clearly inhibited (Supplementary Fig. 13b). Thus, the inhibition site of PMA was suggested to be ferredoxin and/or ferredoxin-NADP(H) reductase FNR as previously reported[36,45]. Besides, we note that P700 oxidation level under illumination with far red (FR) light (i.e., the level before the time point indicated by an upward-pointing arrowhead) was elevated in PMA-treated cells (Supplementary Fig. 12b). The P700 oxidation level under FR light depends on the redox state of PQ governed by the respiration and/or cyclic electron transfer (CET) around PSI[49]; the possibility that PMA acts pleiotropically on CET or respiration cannot be ruled out. Indeed, previous studies suggested that FNR contributes to respiration[50,51]. Explanation of the effect of GA was not straightforward. GA is reported to be a Calvin cycle inhibitor[42] but abolished the slr0688i photocurrent (Fig. 2a). The effects of GA on Y(II) and oxygen-evolving activity were seemingly similar to those of pCMB. However, a closer look at P700 redox kinetics (Supplementary Fig. 12) revealed that the P700 oxidation level under illumination with FR light was elevated in GA-treated cells. This suggests that GA not only inhibits the Calvin cycle but also interferes with respiration- or CET-related cellular metabolic pathway(s). The difference in action between pCMB and GA suggests that electrons involved in photocurrent generation are derived from the components or metabolic pathway(s) that is/are presumably connected to the PQ reduction activity; this issue is discussed in more detail later in this paper. Note that when steady-state NADP(H) contents and the $NADP^+/NADPH$ ratio were quantified, no significant differences between slr0688i and dCas9 were observed (Supplementary Fig. 14). This suggests that the enhanced photocurrent generation from slr0688i is, at least, not associated with the amount of intracellular NADPH.

Finally, we confirmed that, with CP anodes, the effects of the photosynthetic inhibitors on photocurrent generation were basically the same as those observed with ITO anodes (Supplementary Fig. 15). This observation assures that these inhibitors affect the electrogenic activity of the cell itself but not the interaction between the possible mediator(s) and electrode, further validating the above-described results obtained with ITO-based experiments.

**Quantification of EET activity by monitoring ferricyanide reduction**. The photocurrents described above do not necessarily reflect total EET activity; reactivity between unmodified ITO and organic compounds i.e., possible mediator(s), is generally poor[52] and leads to underestimation of EET activity of the cells. Hence, we next used an exogenous electron acceptor, ferricyanide, and evaluated EET more directly at the level of ferricyanide reduction. Cell suspensions of slr0688i and dCas9 were mixed with 1 mM potassium ferricyanide, and changes in the ferricyanide concentration under illumination or in the dark were monitored spectroscopically. Under illumination, slr0688i reduced the extracellularly added ferricyanide at a rate of $2.3 \mu M \ OD^{-1} \ min^{-1}$, whereas dCas9 exhibited no reduction of ferricyanide in the tested

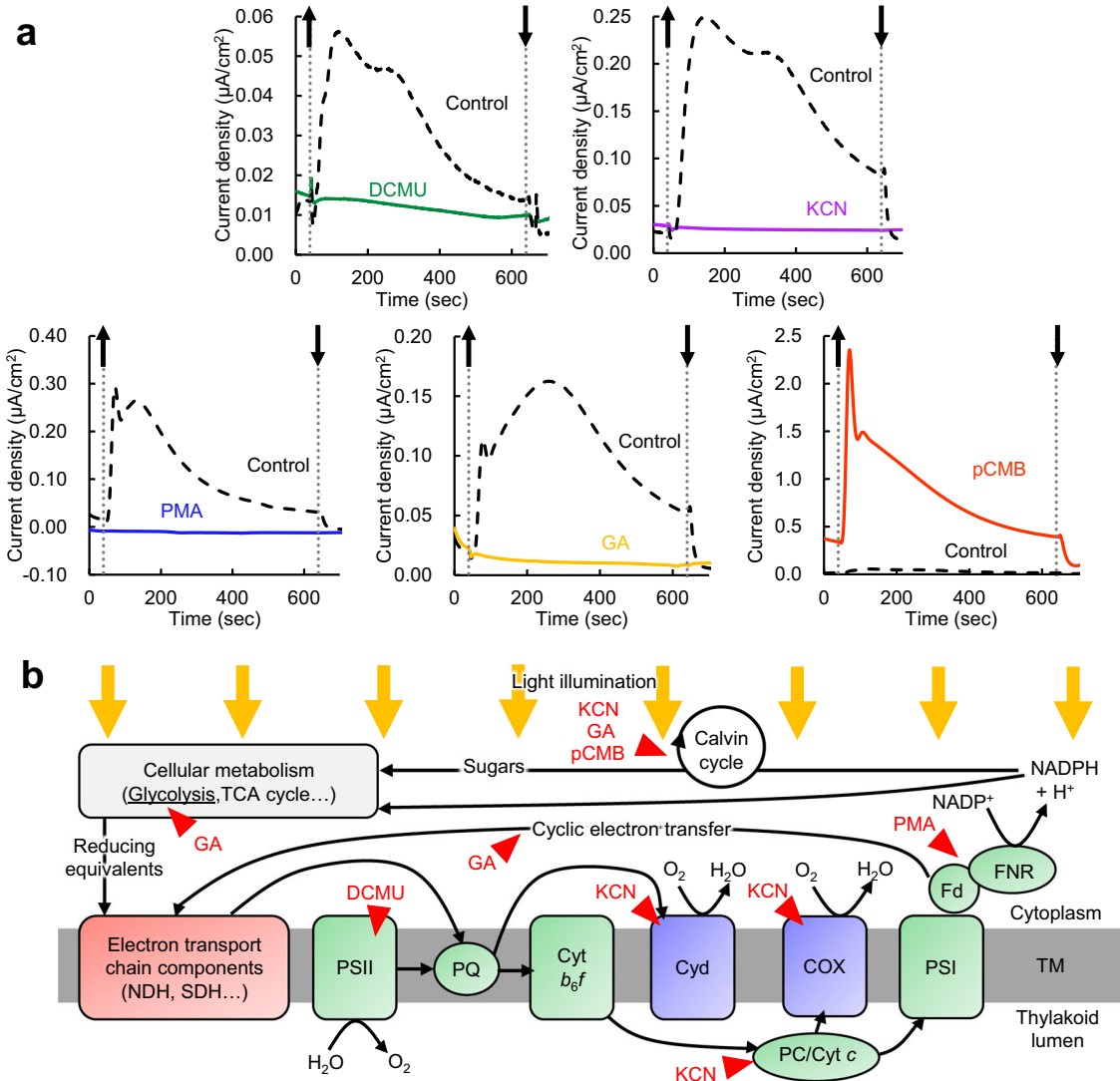

**Fig. 2 Effects of photosynthesis inhibitors on current generation from slr0688i. a** Representative chronoamperograms at +0.25 V vs Ag/AgCl with flat ITO anodes of slr0688i treated with 10 μM DCMU (green line), 5 mM KCN (violet line), 50 μM PMA (blue line), 10 mM GA (yellow line), and 100 μM pCMB (red line) are shown with respective controls (black dashed line). The upward and downward arrows indicate the beginning and end of illumination (70 μmol photons m$^{-2}$ s$^{-1}$), respectively. **b** Target sites of the tested inhibitors. Red triangles show the target sites of the tested inhibitors. The photosynthetic chain of *Synechocystis* cells shares some components with the respiratory chain[86–88]. TM, thylakoid membranes; NDH, NAD(P)H dehydrogenase; SDH, succinate dehydrogenase; Cyd, cytochrome *bd*–quinol oxidase complex; COX, *aa₃*-type cytochrome *c* oxidase; PC, plastocyanin; Cyt *c*, cytochrome *c*. Source data are provided as a Source Data file.

time range (Fig. 3a, c). In the dark, slr0688i reduced ferricyanide at a lower rate (0.8 μM OD$^{-1}$ min$^{-1}$) than under illumination but still at a significantly higher rate than dCas9 in the same conditions (Fig. 3b, d). Substitution of the supernatant with fresh BG11 did not affect the rate of ferricyanide reduction either under illumination or in the dark (Fig. 3), indicating that ferricyanide-mediated EET, which could not be fully detected by the photocurrent measurements (Fig. 1d), was detected by this assay. It has long been considered that ferricyanide freely passes through the outer membrane and accepts electrons from *Synechocystis* cells[53] via some transmembrane protein at the cytoplasmic membrane[11,12]; however, the aforementioned results (Fig. 3) indicated that the outer membrane does serve as a barrier to ferricyanide, which is consistent with its role as a low-permeability molecular sieve that prevents passage of hydrophilic organic molecules[20]. By using inductively coupled plasma mass spectrometry (ICP-MS), we quantified the iron (Fe) concentration in the supernatant of the reaction mixture and confirmed that the total

concentration of Fe did not change during the measurement period (Supplementary Fig. 16). This result indicated that ferricyanide reduction was attributable to the EET activity of the cells and not to the uptake/sequestration of ferricyanide by the cells. Together with the results of the photocurrent measurements, these results clearly showed that deprivation of the outer membrane leads to a significant increase of both mediated and ferricyanide-mediated EET.

**Extracellular derivation of reducing equivalents is enhanced by outer membrane deprivation.** In an attempt to detect and evaluate EET activity more extensively, we examined the electron donation capacity of extracellularly derived reducing equivalents as respiratory substrates for other bacteria. These extracellular reducing equivalents, which are considered energy carriers, i.e. electron donors, in the BPV system[4], cannot be detected and evaluated by photocurrent measurements or a ferricyanide reduction assay. Therefore, *Bacillus cereus* (hereafter *Bacillus*),

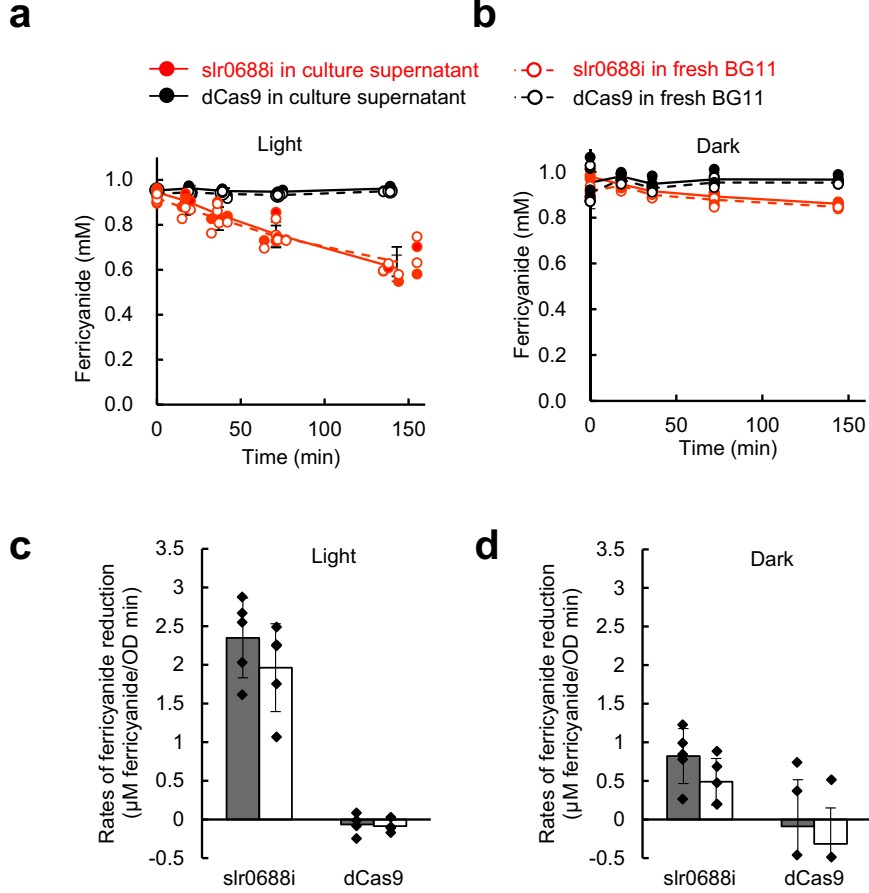

**Fig. 3 Rate of ferricyanide reduction by _Synechocystis_ cells.** Time-dependent reduction of ferricyanide (**a**) under illumination and (**b**) in the dark by slr0688i (red) and dCas9 (black). Each strain was suspended either in their respective culture supernatant (solid lines and closed data point symbols) or in fresh BG11 (dashed lines and open data point symbols) containing 1 mM potassium ferricyanide, and the ferricyanide concentration was measured. Data are presented as mean values ± SD of biologically independent experiments ($n = 4$ for dCas9 under illumination and $n = 5$ for the others). Rates of ferricyanide reduction (**c**) under illumination and (**d**) in the dark by slr0688i and dCas9 suspended in their respective culture supernatants (gray) or in fresh BG11 (white) were then calculated. Data are presented as mean values ± SD of biologically independent experiments ($n = 4$ for dCas9 under illumination and $n = 5$ for the others). Source data are provided as a Source Data file.

which is a gram-positive model organism, was chosen as a recipient of these reducing equivalents. Changes in its membrane potential were monitored as an indicator of electron donation to the respiratory chain because membrane potential generation should be accompanied by extrusion of protons due to electron transfer through the respiratory chain.

Supernatants of slr0688i and dCas9 were added to carbonyl cyanide _m_-chlorophenylhydrazone (CCCP)-treated _Bacillus_ cells and the generation of membrane potential in the _Bacillus_ cells was monitored using the voltage-sensitive dye 3,3′-dipropylthia-dicarbocyanine iodide (DiSC$_3$(5)). To generate enough membrane potential to calculate the rates of formation, the supernatants were concentrated to 5× by ultrafiltration and the obtained retentates (containing compounds with MW > 3000) were added to _Bacillus_ cells. Generation of membrane potential was twice as fast with the supernatant of slr0688i as that of dCas9 (Fig. 4a; Supplementary Fig. 17), indicating that the slr0688i supernatant possessed higher electron donation capacity as respiratory substrates compared with dCas9. Experiments with membrane vesicles isolated from _Bacillus subtilis_ indicated that the compounds included in the supernatant of slr0688i do not donate electrons directly to the respiratory chain of _Bacillus_ cells as a mixture of PMS/ascorbate does (Supplementary Fig. 18); therefore, the observed generation of membrane potential is

suggested to result from metabolization of compounds in the supernatant of slr0688i by _Bacillus_ cells, which feeds electrons to their respiratory chain. Consistent with this, compounds with MW < 3000 were capable of generating membrane potential more than five times as fast as compounds with MW > 3000 (Supplementary Fig. 19), reflecting that simpler, lower molecular weight compounds are metabolized more efficiently and thus donate electrons to the respiratory chain at a faster rate.

To quantify the electron donation capacity as respiratory substrates, the rate of membrane potential generation was measured and compared with those of glucose solutions of different concentrations; a 0.5× concentration of slr0688i supernatant was revealed to generate membrane potential as fast as 4 μM glucose (Fig. 4b; Supplementary Fig. 20); that is, the electron donation capacity of slr0688i supernatant was equal to 8 μM glucose.

## Discussion

In this study, using an outer membrane-deprived _Synechocystis_ mutant, slr0688i[25], we achieved an order-of-magnitude enhancement in photocurrent generation and a significantly higher rate of exogenous ferricyanide reduction. These results verified our hypothesis that the low permeability of the outer membrane contributes to low cyanobacterial EET activity. This

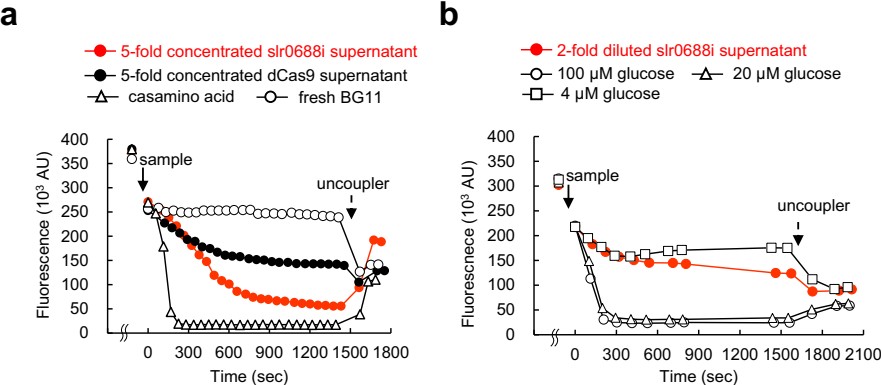

**Fig. 4 Generation of membrane potential of *Bacillus* cells by supernatant of slr0688i.** Changes in membrane potential of *Bacillus* cells were monitored (for detail, see Materials and Methods) following addition of each sample at the time indicated by bold arrows ($t = 0$): (**a**) 5× concentrated supernatants with MW > 3000 of slr0688i (red circles) and dCas9 (black circles), fresh BG11 (white circles), and 0.01% (w/v) casamino acid in BG11 (white triangles), (**b**) a 0.5× concentration of slr0688i supernatant (red circles), 100 µM (white circles), 20 µM (white triangles), 4 µM (white squares) glucose in BG11. At the time indicated by dotted arrows, uncouplers were added to dissipate the membrane potential: (**a**) 5 µM gramicidin, (**b**) 4 µM valinomycin. Source data are provided as a Source Data file.

study demonstrated that an order-of-magnitude increase in current generation from *Synechocystis* cells is achievable by genetic engineering[25]. Moreover, based on the data shown in Fig. 3 and the chlorophyll content of slr0688i (5.6 µg Chl (mL $OD_{730}$)$^{-1}$), the rate of EET from slr0688i to ferricyanide is calculated to be 14.6 nmol electrons nmol Chl$^{-1}$ h$^{-1}$ under illumination; this is higher than the fastest rate reported with terminal oxidase mutants, i.e., 2.7 nmol electrons nmol Chl$^{-1}$ h$^{-1}$ in the Cyd/ARTO double mutant, which we calculated using the data reported in Table S2 in Bradley et al.[12]. The focus of this study was to reveal and characterize the effects of outer membrane deprivation on EET activity of *Synechocystis* cells with regard to current generation, ferricyanide reduction, and electron donation capacity to other bacteria. Therefore, we simply measured the photocurrent generation by the cell suspension that was placed by gravity on the electrode, and did not optimize the electrode or cell–electrode interaction. This study clearly showed that outer membrane deprivation is a promising technology to improve the efficiency of BPV systems. The photocurrent is expected to be further enhanced by combining various other approaches: optimization[54] and modification of the electrodes[14,33,34], strengthening the cell–electrode interaction[13,14,30,32,55], optimizing the cell culture conditions[56], and genetic engineering[12,31] in addition to outer membrane deprivation. Indeed, as demonstrated in Fig. 1e, photocurrent generated from slr0688i could be increased more than 100-fold by using larger amount of the cells and placing carbon paper onto ITO as an additional anode; the observed photocurrent is the highest among those reported for mediator-less setups utilizing untreated whole cells of *Synechocystis*[12–14,31–34]. When using a carbon paper anode, we noticed that the current level tended to remain high after switching off the light (Fig. 1e), and it took approx. 1 h to decline back to the basal level. We speculate that this is probably because pores of carbon paper (about 100 µm)[57] prevent mediators reduced by cells from diffusive dispersion into the bulk electrolyte and/or trap cells so densely that sufficient $CO_2$ is not available, leading to the overaccumulation of intracellular reducing equivalents which could possibly reduce mediators even in the dark.

This study provides the useful means for EET enhancement by structural alteration of a genetically manipulated cell surface. The outer membrane of the slr0688i mutant is detached from the peptidoglycan layer by suppression of pyruvylation of peptidoglycan-linked polysaccharides (Supplementary Fig. 1a),

which weakens the interaction between peptidoglycans and the S-layer homologous (SLH) domain of outer membrane proteins[25]. This leads to liberation of periplasmic and thylakoid lumenal components[25], which were proven in this study to have an electron donation capacity equal to 8 µM glucose (Fig. 4b). Although slr0688i exhibits growth retardation compared to the wild type (Supplementary Fig. 1b), photosynthetic and respiratory activity were comparable to those of the wild type[25] as judged by the oxygen-evolving activity (Supplementary Fig. 2).

The importance of cyanobacterial cell membranes on the photocurrent generation was previously demonstrated by Saper et al., where *Synechocystis* cells treated either through a low-pressure microfluidizer at 10–15 psi under the presence of 400 mM NaCl or by a simple osmotic shock using 400 mM NaCl produced markedly enhanced photocurrents compared to the untreated cells[15]. These observations show that a gentle physical treatment, which presumably causes a modest damage(s) to the cell membranes, leads to a higher electrogenic activity and thus their study provided a notable approach to improve the bio-photoelectrochemical system. However, although both the 'physically-treated' cells and the outer membrane-deprived cells presented by our study show the enhanced electrogenic activity, their underlying mechanisms are considered different from each other for the following reasons. First, aforementioned physical treatments might partially perturb the structural integrity of the cell surface but is not expected to deprive the outer membrane from the cells because the cyanobacterial outer membrane is well known to be tightly anchored to the peptidoglycan even after the mechanical disruption of the cells, such as passing through a French pressure cell press at 14,000 psi[20] or mechanical cracking[58,59]. Detaching the outer membrane thus requires the enzymatic degradation of the peptidoglycan and/or solubilization of the outer membrane by detergent treatment[20,58–60]. Second, suspending the cells in 400 mM NaCl unequivocally generates the osmotic pressure to the cytoplasmic membrane rather than to the outer membrane because the outer membrane contains abundant non-specific channels those allow the rapid passage of ions across the outer membrane[20]. Indeed, Reed et al.[61] showed that treating the cells with 490 mM NaCl affected the cytoplasmic membrane and resulted in the leakage of various cytosolic metabolites, which may include the possible mediator(s) for EET. Third, genetically engineered outer membrane deprivation avoids interfering the cytoplasmic membrane function and cytosolic metabolites release, so that even abundant cytosolic molecules, such as

NADP(H), was not detected in the culture supernatant of the outer membrane-deprived cells[25] (Supplementary Fig. 21, which is discussed in detail later in Discussion section). Consequently, it appears reasonable to assume that the electron transfer path from the physically-treated cells and that from the outer membrane-deprived cells, to the electrodes, are different; as a matter of fact, this assumption was verified by a striking difference in their photocurrent properties that DCMU enhances the photocurrent of the former cells[15] but abolishes the photocurrent from the latter cells (Fig. 2a).

Photocurrent measurements under the presence of a number of photosynthesis inhibitors demonstrated that the electrons stem from the pathway(s) downstream of PSI (Fig. 2, Supplementary Figs. 9–13). Among the effects of the inhibitors, the difference in effects of GA and pCMB appeared to be important; both compounds are suggested to inhibit the Calvin cycle (Supplementary Figs. 10–13)[39,42], but the former inhibitor abolished the photocurrent, whereas the latter markedly enhanced it (Fig. 2a). One obvious difference in effects between GA and pCMB was that only GA-treated cells exhibited a high P700 oxidation level under FR light illumination (Supplementary Fig. 12), suggesting that GA inhibits respiration- and/or CET-related pathway(s). It is not unreasonable to assume the inhibitors affect pathways other than the Calvin cycle because GA is known to be inherently pleiotropic owing to its electrophilicity; indeed, GA is reported to inhibit yeast growth and fermentation[62]. An additional difference between GA and pCMB was evident from NADPH fluorescence measurement (Supplementary Fig. 13); the pCMB-treated cells showed a slowed decay of NADPH fluorescence after switching off the light and NADPH formation in the subsequent dark period was never observed. The slowed NADPH fluorescence decay may be attributable to Calvin cycle inhibition, but the lack of subsequent NADPH formation in the dark suggests the existence of other inhibitory effects of pCMB on the metabolic pathway(s) that reduces NADP, such as the oxidative pentose phosphate pathway[63,64]. We speculate that inhibition of dark NADPH formation changes cellular metabolism to accumulate alternative reducing equivalents instead of NADPH and this may be linked to the enhanced EET activity. Supporting this speculation, the pCMB-treated cells showed greatly enhanced EET current even in the dark (see the amperometric i-t curve in Fig. 2a). Taken together, the electrons involved in photocurrent generation seem to be derived from intermediates of metabolic pathway(s) that is linked to PQ reduction but not to NADP reduction. Elucidation of the action of GA and pCMB will provide further insights into the EET mechanism.

The notion that the photocurrent stems from the pathway(s) downstream of PSI is largely consistent with previous studies[2,11,12,17]. In addition, the possible involvement of the pathway(s) connected to PQ reduction agreed with the previous observation that a mutant deficient in the functional NDH complex, which supplies electrons to PQ, exhibits significant enhancement of ferricyanide reduction[12]. However, contradictory observations hinder a straightforward understanding of the EET mechanism of this cyanobacterium. (I) Strikingly, some studies reported that DCMU enhances photocurrent generation[15,17], whereas others, including this study, reported the opposite effect[11,13,14]. (II) Recently, Shlosberg et al. reported that NADPH serves as the electron mediator in this bacterium[17]. However, this mechanism is not necessarily applicable to this study because of the following two reasons. (I) We confirmed that the supernatant of slr0688i cells, regardless of before and after the electrochemical measurement, contained no detectable level of NADPH that was analyzed by means of the two-dimensional fluorescence mapping (2D-FM), enzymatic analysis, and LC-MS/MS analysis (Supplementary Fig. 21). The 2D-FL revealed the presence of strong

fluorescent molecules and one of which exhibited the fluorescence pattern of excitation wavelength (Ex) of 360 nm and emission wavelength (Em) of 450 nm, which was similar to those of NADPH (Ex 340 nm/ Em 460 nm). However, the small discrepancy of Ex/Em led us to further analyze whether this fluorescence was actually derived from NADPH; neither an enzymatic assay whose detection limit was 0.0625 μM nor a LC-MS/MS assay using the 50-fold concentrated supernatant detected NADPH. On the basis of these results, we concluded that the fluorescent molecules found in the supernatant was not NADPH. The absence of NADPH in the supernatant is reasonable because the bacterial periplasm is well known to be an oxidizing environment, and bacteria possess various systems to convey reducing power to periplasmic space rather than directly exporting the cytosolic reducing equivalents such as NAD(P)H (one of the examples can be found in Depuydt et al.[65]). Thus, just depriving the outer membrane is not considered to result in releasing NADPH into the external environment. (II) Furthermore, intracellular NADPH fluorescence intensity was not significantly changed even in pCMB-treated cells despite their showing greatly enhanced photocurrent generation (Fig. 2a, Supplementary Fig. 13a). One possible explanation for these contradictory observations is the assumption that the EET pathway may depend on the cellular physiology and intracellular electron flux, which are sensitive to numerous growth condition parameters, such as light intensity or $CO_2$ availability. Thus, further investigation is necessary to elucidate the EET mechanism of this organism[66–68].

Deprivation of the outer membrane is undoubtedly beneficial not only for the improvement of BPV systems but also for enhancing production of various chemicals ranging from biofuels to high-value compounds[69–73]. Supporting this is a previous study showing that slr0688i secretes enough nutrients to support the growth of heterotrophs[25]; moreover, this study also showed that the supernatant of slr0688i provides respiratory substrates to *Bacillus* cells as efficiently as 8 μM glucose. Many beneficial aspects of the outer membrane-deprived mutant remain to be explored, and full utilization of its photosynthetic reaction is a promising way to achieve a clean and sustainable future.

## Methods

**Bacterial strains and general growth conditions**. The *Synechocystis* strains used in this study were slr0688i, whose outer membrane is deprived due to conditional repression of *slr0688* by the CRISPRi system regulated by both dCas9 and sgRNA, and dCas9, a control strain (serving as the wild-type strain in this study) whose chromosomal DNA contains the same genetic construct as slr0688i without the sgRNA[25]. These strains were grown photoautotrophically at 30 °C, 100 μmol photons m$^{-2}$ s$^{-1}$, with constant shaking at 140 rpm in BG11 medium consisting of the following ingredients (per L): 2 mL of Solution I (0.5 g/L Na$_2$EDTA·2H$_2$O, 3 g/L ammonium iron (III) citrate, 3 g/L citric acid), 25 mL of Solution II-a (60 g/L NaNO$_3$, 3 g/L MgSO$_4$), 25 mL of Solution II-b (1.56 g/L K$_2$HPO$_4$), 2 mL of Solution III (14.3 g/L CaCl$_2$), 1 mL of A6 Solution (2.86 g/L H$_3$BO$_3$, 1.81 g/L MnCl$_2$·4H$_2$O, 0.22 g/L ZnSO$_4$·7H$_2$O, 0.08 g/L CuSO$_4$·5H$_2$O, 0.021 g/L Na$_2$MoO$_4$·H$_2$O, 0.0494 g/L Co(NO$_3$)$_2$·6H$_2$O, 1 droplet of H$_2$SO$_4$), and 20 mL of 1 M TES-KOH (pH 7.5). For cultivation, 100 to 125 mL volumetric flasks containing 12 mL of culture were generally used. Deprivation of the outer membrane of slr0688i was induced by treating precultured cells dilluted to OD$_{730}$ = 0.1 with 1 μg/mL anhydrotetracycline dissolved in dimethyl sulfoxide (DMSO), as described in Kojima and Okumura[25]. *Synechocystis* cells at log phase were used in all the following experiments.

**Preparation of stock solutions of photosynthesis inhibitors**. DCMU, pCMB and PMA were purchased from Tokyo Chemical Industry Co., Ltd. (Tokyo, Japan). KCN was purchased from FUJIFILM Wako Pure Chemical Corporation (Osaka, Japan). GA dimer was purchased from Sigma–Aldrich (St. Louis, MO, USA). DCMU, pCMB and PMA were dissolved in DMSO; KCN and GA were dissolved in water. Concentrations of stock solutions were as follows: 10 mM for DCMU, 100 mM for pCMB, 50 mM for PMA, 5 M for KCN and 1 M for GA.

**Electrochemical measurements**. The electrochemical setup was composed of a cylindrical glass chamber (Φ 20 × 30 mm; geometrical surface area, 3.14 cm$^2$), an ITO-coated glass (GEOMATEC) as a working electrode placed at the bottom of the chamber, a platinum wire as a counter electrode, and a Ag/AgCl reference electrode

(HOKUTO DENKO). All electrochemical measurements were conducted at 30 °C with either a potentiostat/galvanostat HA-1510 (HOKUTO DENKO) or an electrochemical analyzer Model 760 C (ALS). Data collection was performed with GL220_820APS Version 1.12 and ALS/CHI electrochemical analyzer Version 14.01 J. Data analysis was performed with Microsoft Excel for Microsoft 365 MSO Version 2202.

Common procedures for electrochemical measurements performed in this study were as follows: Synechocystis cell suspensions after 4 days of culture (unless otherwise stated) were first diluted to $OD_{730} = 1.5$, 4 mL of which were then separated into sedimented cells and supernatant by centrifugation at $2500 \times g$ for 10 min, and the supernatant was first injected into the electrochemical chamber. The sedimented cells were then resuspended with the rest of the supernatant, drawn up into a syringe, injected by gravity onto the ITO glass at the bottom of the chamber, and incubated to settle the cells down. The approximate amounts of cells used in electrochemical experiments were $6.0 \times 10^8$ cells (33.6 µg Chl) and $2.4 \times 10^9$ cells (27.6 µg Chl) for slr0688i and dCas9, respectively (cell number and chlorophyll contents were as follows: $OD_{730} = 1 \times 10^8$ cells mL$^{-1}$ and $5.6 \pm 0.3$ µg Chl (mL $OD_{730}$)$^{-1}$ for slr0688i, and $OD_{730} = 4 \times 10^8$ cells mL$^{-1}$ and $4.6 \pm 0.2$ µg Chl (mL $OD_{730}$)$^{-1}$ for dCas9). Typical photographs and microscopic images of the cells residing on the ITO surface are shown in Supplementary Fig. 22. Data collection was performed with FV10-ASW Version 04.02; data analysis was performed with FV10-ASW Version 04.02 and OriginPro 2021b (64-bit) SR2 Version 9.8.5.212.

Prior to electrochemical measurements, pieces of carbon paper (TGP-H-060, Toray) were submerged in a 3:1 (v/v) mixture of $H_2SO_4/HNO_3$ and incubated at 80 °C for 2 h; this treatment hydrophilized carbon papers by COOH formation[74,75]. Note that carbon paper without any treatment is not suitable for BPV operation[54]. This carbon paper was placed onto ITO (projected surface area, 3.14 cm$^2$), and the electrochemical measurements were performed likewise as described above.

To avoid detection of a pseudo-photocurrent, which is explained in detail below, the pH of Synechocystis cell suspensions was always confirmed and adjusted when necessary prior to electrochemical measurements. The pH of the suspension preferably should be below approximately 8.0, and should never be above 8.5. Otherwise, the buffering capacity of TES (pH 6.8–8.2) is outcompeted by drastic changes (increases) in the pH of the medium due to photosynthesis[76–78], resulting in detection of a pseudo-photocurrent of the following reaction:

$$Mn^{2+} + 2H_2O \rightleftharpoons MnOOH + 3H^+ + e^- \qquad (1)$$

where $Mn^{2+}$ comes from $MnSO_4$, a component of the BG11 medium. Some examples of detected pseudo-photocurrents of manganese are shown and described in detail in Supplementary Fig. 23. Therefore, before electrochemical measurements, the pH of cell suspensions was always confirmed to be below approximately 8.0, and when necessary, fresh BG11 or HCl was added not only to adjust the optical density, but also the pH of cell suspensions below approx. 8.0. In chronoamperometry, the applied voltage was fixed to +0.25 V vs. Ag/AgCl, at which sufficient current generation occurred (Supplementary Fig. 24) and a psedo-photocurrent was never detected under an appropriate pH (Supplementary Fig. 23).

When the supernatant needed to be replaced, it was removed after centrifugation and substituted with the supernatant of interest, which was prepared by filtration with an Ultrafree-MC-GV, 0.22 µm (Millipore), or with fresh BG11.

Size-fractionated supernatants were prepared as follows: the supernatant was first filtrated from a cell suspension of slr0688i with Millex-GV Syringe Filter Unit, 0.22 µm (Millipore) and freeze-dried with a freeze dryer (TOKYO RIKAKIKAI). The obtained powder was suspended with water to yield a 10× concentrated supernatant and size fractionated by ultrafiltration with Amicon Ultra Centrifugal Filters (Millipore) to yield a supernatant containing compounds of MW > 50,000, >30,000, >3000 and <3000. Each fraction was then diluted to 0.1× concentration with fresh BG11, and mixed with slr0688i cells to measure photocurrent generation.

For experiments using pCMB, DCMU, and KCN, cell suspensions adjusted to $OD_{730} = 1.5$ and pH 7.6 by adding fresh BG11 were incubated with 100 µM pCMB, 10 µM DCMU, or 5 mM KCN for 1.5 h in the dark on ITO in the electrochemical chambers prior to electrochemical measurements. Experiments using 50 µM PMA and 10 mM GA were performed with 4 mL of cell suspensions of $OD_{730} = 1.5$, followed by incubation for approx. 1 h in the dark on ITO prior to electrochemical measurements. For the control measurements, the respective volume of solvents, i.e., water or DMSO, were added to cell suspensions and incubated likewise as described above.

**Microbial adhesion to hydrocarbons assay.** Following the previous report[26], Synechocystis cell suspensions after 3 days of culture were washed and adjusted to $OD_{730} = 2.0$ with 1.2 mL BG11, followed by addition of 200 µL p-xylene, and then vortexed for 120 s. After 15 min, the aqueous phase was collected and absorbance at 730 nm was measured with a UV/VIS spectrophotometer UV-1850 (SHIMADZU). The percentage of the cells adsorbed by hydrocarbon (%) was calculated as [{($A_{730}$ before p-xylene addition) − ($A_{730}$ of aqueous phase after mixed with p-xylene)} × 100] / ($A_{730}$ before p-xylene addition).

**Ferricyanide reduction assay.** The rate of ferricyanide reduction was measured using 2.1 mL of Synechocystis cell suspensions after 3 or 4 days of culture adjusted

to $OD_{730} = 1.0$. Then, 1 mM potassium ferricyanide was added to the cell suspensions and they were incubated at 30 °C with constant shaking at 140 rpm either under illumination (50 µmol photons m$^{-2}$ s$^{-1}$) or in the dark. Changes in the concentration of ferricyanide (extinction coefficient, 1.052 mM cm$^{-1}$) were monitored up to approx. 140 min by measuring absorbance at 420 nm with a UV/VIS spectrophotometer UV-1850 (SHIMADZU).

The Fe concentration was measured by ICP-MS (Agilent 7700) using a dilution series of ferricyanide solution as standards. Synechocystis cell suspensions were collected and washed twice, and resuspended to $OD_{730} = 1.0$ with BG11 not containing $Na_2EDTA$ and ammonium iron (III) citrate (Fe-free BG11). Then, 1 mM ferricyanide was added to 2.1 mL of the cell suspensions, which were incubated at 30 °C with constant shaking at 140 rpm under illumination (50 µmol photons m$^{-2}$ s$^{-1}$). Before and after incubation for 2 h, the concentration of ferricyanide and Fe in filtrated supernatants were measured with a spectrophotometer and ICP-MS, respectively.

**Chlorophyll fluorescence measurements.** Chlorophyll fluorescence was measured with a PAM-2500 (Walz) instrument. Slr0688i cells collected and resuspended with fresh BG11 ($OD_{730} = 13$) were incubated with 100 µM DCMU, 5 mM KCN, 50 µM PMA, or 10 mM GA for 10 min, followed by chlorophyll fluorescence measurement. In the case of pCMB, cells ($OD_{730} = 1.5$) were preincubated with 100 µM pCMB in the dark, followed by centrifugation and resuspension ($OD_{730} = 13$); the total duration of pCMB treatment was 1.5 h. Control samples were prepared by incubating cells with the respective solvents for the same time period. A Suspension Cuvette (KS-2500, Walz) with a magnetic stirrer was used for all measurements. The measuring light was set at an intensity of 4 with PamWin-3 software. The measurement frequency was programmed to increase from 500 to 20,000 Hz during illumination with red actinic light of 630 nm at 106 µmol photons m$^{-2}$ s$^{-1}$. Saturating flashes (600 ms, at an intensity of 10 set with PamWin-3 software) were applied every 20 s during actinic light illumination to obtain the maximum fluorescence of light-adapted cells ($F_m'$). The effective quantum yield of PSII, Y(II), was calculated as ($F_m' - F_s$) / $F_m'$, where $F_s$ is the steady-state fluorescence obtained during actinic light illumination. Data collection was performed with PamWin Version 3.22d; data analysis was performed with Microsoft Excel for Microsoft 365 MSO Version 2202.

**Steady-state oxygen evolution and uptake measurements.** The photosynthetic activity of Synechocystis cells was evaluated with a Clark-type electrode (Hansatech) at 25 °C. To measure steady state photosynthetic activity, 2 mL of cell suspension (12 µg Chl mL$^{-1}$) containing 5 mM $NaHCO_3$ was illuminated with a CoolLED pE-100 LED either at 750 or 120 µmol photons m$^{-2}$ s$^{-1}$. Dark respiration was evaluated from the oxygen consumption rate before switching on the light, and photosynthetic activity was calculated by adding the rate of dark respiration to that of oxygen evolution under illumination.

The same oxygen electrode setup was used to analyze the effects of photosynthesis inhibitors on photosynthetic activity. In the case of KCN, Synechocystis cells were collected and resuspended with fresh BG11 to a final concentration of 12 µg Chl mL$^{-1}$, and mixed with 5 mM $NaHCO_3$ and 5 mM KCN, immediately after which oxygen evolving activity was measured. In the case of the other inhibitors, Synechocystis cells collected and resuspended with fresh BG11 (24 µg Chl mL$^{-1}$) were incubated with 100 µM pCMB for 1.5 h, 50 µM PMA for 0.5 h or 10 mM GA for 0.5 h in the dark. Control samples were prepared by incubating cells with the respective solvents for the same time period. The cell suspensions were then mixed with 5 mM $NaHCO_3$ and diluted to 12 µg Chl mL$^{-1}$ with fresh BG11, followed by immediate measurement with an oxygen electrode. Data collection was performed with OxyTrace+ Windows software Version 1.0.48; data analysis was performed with Microsoft Excel for Microsoft 365 MSO Version 2202.

**NADPH fluorescence measurement.** NADPH fluorescence was measured with a DUAL-PAM-100 (Walz) instrument[79,80]. Slr0688i cells collected and resuspended with fresh BG11 (24 µg Chl mL$^{-1}$) were incubated with 100 µM pCMB for 1.5 h, 50 µM PMA for 0.5 h, or 10 mM GA for 0.5 h in the dark. Control samples were prepared by incubating cells with the respective solvents for the same time period. The cell suspensions were then injected and diluted to 2.4 µg Chl mL$^{-1}$ in a $1 \times 1$ cm cuvette installed in the DUAL-PAM. The measuring light of 365 nm was set at an intensity of 10 on the DUAL-PAM software, and the measuring frequency was programmed to increase from 50 to 500 Hz during illumination with actinic light of 635 nm at 1090 µmol photons m$^{-2}$ s$^{-1}$. Upon illumination with actinic light, no changes in fluorescence were observed with BG11 medium supplemented only with the inhibitors, i.e., without Synechocystis cells. Data collection was performed with Software Dual PAM Version 3.12(Windows 10); data analysis was performed with Microsoft Excel for Microsoft 365 MSO Version 2202.

**P700 measurements.** Slr0688i cells collected and resuspended with fresh BG11 (24 µg Chl mL$^{-1}$) were incubated with 100 µM pCMB for 1.5 h, 50 µM PMA for 0.5 h, or 10 mM GA for 0.5 h in the dark. Control samples were prepared by incubating cells with the respective solvents for the same time period. The samples were then injected into a $1 \times 1$ cm cuvette, and P700 absorption changes were

measured with a fiber version DUAL-PAM/F (Walz) instrument, with the surface of the fiberoptics completely attached to the side of the cuvette. Measurements were performed in the remission mode, where measuring light (830/875 nm, intensity 10 with the DualPAM software) from the fiberoptics entered the cuvette and, after scattering and partial absorption by the cell suspension, was picked up by the fiberoptics from the same side. Changes in P700 absorption were measured by applying the multiple-turnover flash (300 ms, intensity of 16 with the software) following illumination for 10 s with FR light (720 nm, 2.7 μmol photons m$^{-2}$ s$^{-1}$, which was an intensity of 20 with the DualPAM software). Data collection was performed with Software Dual PAM Version 3.12(Windows 10); data analysis was performed with Microsoft Excel for Microsoft 365 MSO Version 2202.

**NADP$^+$/NADPH enzymatic assay**. The amount of NADP$^+$/NADPH was quantified using an NADP$^+$/NADPH Assay Kit-WST (DOJINDO), following the manufacturer's instruction. The extraction of intracellular NADP(H) of *Synechocystis* cells[81] was performed as follows: cells were collected centrifugally and resuspended in TE saturated phenol (NIPPON GENECO., LTD), followed by the addition of the same volume of Extraction Buffer supplied in the kit, vigorous mixing and centrifugation at $17,800 \times g$ for 5 min. The obtained supernatant was transferred to a new tube and the same volume of chloroform was added. After vigorous mixing and centrifugation, supernatant was obtained and used for quantification of NADP(H).

**Fluorometric measurement of membrane potential of *Bacillus* cells**. *Bacillus cereus* (hereafter *Bacillus*) cells were cultured at 30 °C with constant shaking at 140 rpm in Luria-Bertani (LB) medium. An overnight preculture was diluted 100-fold in fresh LB medium and grown for another 5 h before analysis. *Bacillus* cells were then collected by centrifugation at $10,000 \times g$ for 4 min and resuspended in fresh BG11 medium, whose OD$_{600}$ was adjusted to approx. 0.3. CCCP dissolved in DMSO was added to a final concentration of 30 μM, and the cell suspension was incubated for 30 min at 30 °C with constant shaking at 140 rpm. The cells were then collected and washed once with BG11 to remove the CCCP, followed by final resuspension with BG11, in which the cells were allowed to rest for about 40 min (here, the OD$_{600}$ of the cell suspension was supposed to be approx. 0.6). A voltage-sensitive dye, DiSC$_3$(5), was used to monitor changes in the membrane potential, following Winkel et al.[82]; depolarization of the membrane is reflected as an increase in fluorescence originating from the dye, whereas repolarization leads to a decrease in fluorescence[83]. Initial fluorescence levels were first recorded with the cell suspension incubated with 2 μM DiSC$_3$(5) for about 5 min prior to the measurement. After the addition of supernatant of either slr0688i or dCas9 of the same volume as the cell suspension, making the concentration of DiSC$_3$(5) 1 μM, the fluorescence emission spectra of the cell suspension, excited at 610 nm, were recorded from 630 to 700 nm time-dependently with an FP-8500 spectrofluorometer (JASCO). Changes in the fluorescence intensity at 674 nm were plottted to analyze changes in membrane potential across the cytoplasmic membrane of *Bacillus* cells. Data collection was performed with Spectra Manager Version 2.12.00; data analysis was performed with Microsoft Excel for Microsoft 365 MSO Version 2202.

Casamino acid and glucose dissolved in fresh BG11 were used as positive controls, and fresh BG11 was used as a negative control. Supernatants were prepared by filtration from cell suspensions of slr0688i or dCas9 after 3 days of culture with Millex-GV Syringe Filter Unit, 0.22 μm (Millipore). Concentration and size fractionation were done by filtration with Amicon Ultra Centrifugal Filters (Millipore). All the supernatants and control samples were confirmed to be at the same pH (within approx. pH 7.6–7.8) before the measurements, and when necessary, were adjusted to pH 7.6–7.8 by addition of HCl. Membrane vesicles free of intracellular components were prepared from *Bacillus subtilis* according to the method described in Konings et al.[84]. Note that because the supernatant of slr0688i contained phycocyanins[25], it yielded unignorable fluorescence in the tested range from 630 to 700 nm when excited at 610 nm (Supplementary Fig. 25), which did not interefere with changes in fluorescence from DiSC$_3$(5). The fluorescence from phycocyanins disappeared when purged with oxygen, which leads to self-sensitized bleaching of phycocyanins[85], or treated with heat (Supplementary Fig. 25).

**Electron-microscopy**. Cells after 3 days of culture were harvested by centrifugation at $2000 \times g$ for 10 min at 4 °C. The cells were then fixed with 2% glutaraldehyde and were ultra thin-sectioned[25]. The thin-sectioned samples were stained with 2% uranyl acetate and 0.4% lead citrate, and observed using a transmission electron microscope (JEM-1400Plus; JOEL Ltd, Tokyo, Japan) at an acceleration voltage of 100 kV.

**Two-demensional fluorescence map measurements**. Supernatants of slr0688i cells were obtained by centrifugally removing cells from culture, followed by filtration with Millex-GV Syringe Filter Unit, 0.22 μm (Millipore). Fluorescence spectra of the supernatants were measured using FP-8500 spectrofluorometer (JASCO, Tokyo, Japan). The excitation wavelength was set to 260 nm to 450 nm and the emission wavelength was set to 275 nm to 700 nm. Data collection was performed with Spectra Manager Version 2.12.00; data analysis was performed with OriginPro 2021b (64-bit) SR2 Version 9.8.5.212.

**Liquid chromatograph-mass spectrometry analysis**. Supernatants of slr0688i cells were collected by centrifugation at $2500 \times g$ for 10 min at room temperature. Ten milliliter of the supernatant was filtered through a 0.22 μm pore size polyvinylidene difluoride membrane, then frozen by liquid N$_2$, and lyophilized. The lyophilized material was dissolved in 200 μL of distilled water. Ten microliter of the dissolved sample was mixed with 10 μL of acetonitrile and 180 μL of 10 mM borate-NaOH buffer at pH 8.7, and analyzed by LCMS-IT-TOF (Shimadzu, Kyoto, Japan). The standard NADPH-Na was purchased from Nacalai Tesque, Inc (Kyoto, Japan), and 10 μL of 25 μM NADPH solution was mixed with acetonitrile and borate-NaOH buffer as described above. The LC settings were as follows: Column, TSKgel ODS-100V (Tosoh, Tokyo, Japan); flow rate, 0.3 mL/min; eluent A, 25 mM ammonium formate at pH 6.0; eluent B, 60% acetonitrile. The timetable of the eluent A/B gradient was as follows: 0 min, 3%; 3.70 min, 5%; 3.71 min, 12%; 6.95 min, 30%; 7.95 min, 60%; 7.96 min, 95%. Mass-spectrometry measurements were performed with electrospray ionization in positive ion mode, under dissociation line temperature at 200 °C. Mass spectra were acquired in the mass rage of 100–1000 m/z. Data collection was performed with LCMSsolution Version 3.80; data analysis was performed with Microsoft Excel for Microsoft 365 MSO Version 2202.

**Confocal laser scanning microscopy imaging**. Cells after 4 days of culture were dropped onto an ITO electrode and covered with a cover glass so that the number of the cells per surface area is approximately the same as that in the case of electrochemical measurements performed with ITO electrode. The imaging of autofluorescence from the cells was performed with FV1000/BX61 confocal laser scanning microscope (Olympus, Japan) by utilizing Cy5-filter sets. Data collection was performed with FV10-ASW Version 04.02; data analysis was performed with FV10-ASW Version 04.02 and OriginPro 2021b (64-bit) SR2 Version 9.8.5.212.

**Reporting summary**. Further information on research design is available in the Nature Research Reporting Summary linked to this article.

## Data availability
All relevant data supporting the key findings of this study are available within the article and its Supplementary Information files. Source data are provided with this paper.

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

## Acknowledgements

We thank Yasuhiro Hashimoto (Panasonic Corp.) for assistance with measurement of the Fe concentration in supernatants by ICP-MS. This study was supported by Panasonic corp.

## Author contributions

S.Ku., S.Ko. and S.N. conceived the idea. S.Ku., S.Ko., G.S., C.M., K.T. and S.N. designed the experiments. S.Ku. and S.Ko performed the experiments. S.Ku., S.Ko., K.K., G.S., C.M., K.T., Y.O. and S.N. analyzed the data. S.Ku., S.Ko, G.S, C.M, K.T and S.N wrote the paper. S.Ko. and S.N. supervised the entire research.

## Competing interests

Authors S.Ku., S.Ko., and Y.O. are employed by Panasonic corp. Other authors declare no competing interests.
