## [Peer Review File · Nature Communications]

Order-of-magnitude enhancement in photocurrent generation of *Synechocystis* sp. PCC 6803 by outer membrane deprivationReviewers' Comments:

Reviewer #1:

Remarks to the Author:

Reviewing the manuscript by Kusama et al " A 35 fold" submitted to Nature comm.

The authors used a formally generated mutant of *Synechocystis*, slr0688i, in which the cell membrane is deprived for the entrance and moving out of metabolites and proteins. Using this mutant in a biophotovoltaic cell generated 35 fold enhancement in the EET current. The study of the EET phenomenon in bacteria, cyanobacteria and algae is intensive and important in order to be able to generate an electricity producing photo-cell that will make use of live microorganisms and light to generate green energy. However, several points prevent this manuscript of being a novel one and providing new addition to the field.

1. The currents produced in the photoelectric cell are significantly lower than the current produced by cyanobacteria in other publications. This is true even to the enhanced currents generated by the mutant in this work. This is probably since an ITO anode was used. It is already well established that the carbon-graphite electrode generates more than x10 higher currents.

2. It was already shown that manipulating the cell membrane by physically means like an osmotic treatment or low-pressure microfluidaizer treatment significantly enhances the EET in *Synechocystis* (ref 15). Therefore, this study repeats this phenomenon in a genetic eng manipulated technique.

3. It has been recently reported that the electron mediator in this organism is NAPH (ref 17).

Therefore, the following work should study this component as the electricity generated component.

Minor notes:

1. The nature of the membrane deprived mutant sl0688i is not describe and the reader should go to a manuscript in bioArxiv (ref 25) to study the details. It should be described in the present manuscript in details and the molecular mechanism discussed in the discussion. Perhaps it would be better and naturally to combine the two manuscripts.

2. Line 41. What is "higher photosynthetic activity than plants". In what sense?

3. Figure 1. This is a nice schematic drawing but add no result to the work. Could be displayed as a panel in another figure.

4. Figure 2. As said above, the phot-currents are significantly low compared to what already achieved with live *Synechocystis*.

5. Figure 3. DCMU has been shown in other work to increase the photo-current generated by live *Synechocystis*. Discuss what is the possible difference.

Reviewer #2:

Remarks to the Author:

Dear researchers of that paper!

First I want to tell you that I was very excited by the topic. Since researchers always have to prove the impact of their research onto the society, I think this paper will help to even further increase the economic importance of cyanobacteria (after approaches to use them for the gain of biofuel within the last decade). As my topic in the past has been heterotrophy and transport across the cytoplasmic membrane, I am clearly impressed first about the deprivation of a cyanobacterium's outer membrane for the first time by your group in a separate work and second the increased external electron transfer by the same mutant strain discovered in this work. So over all there is hardly anything I can criticize. However, I would be interested, how long it takes after resuspension in fresh BG11 medium to establish photocurrent again after enough redox active compounds have been secreted, since your cells have also been inoculated in fresh medium at the beginning of cultivation before the actual experiment. You mentioned that KCN inhibits all respiratory terminal oxidases but also plastocyanine (lines 148-149). However, cytochrome c can also serve as an electron carrier in photosynthesis thereby completely replacing plastocyanine in its function since plastocyanine deletions have already been performed in *Synechocystis* sp. PCC 6803 (Pils and Schmetterer, 2001, FEMS Microbiol Lett 203, 217-222). In Fig. 3b cytochrome c should also be displayed. You write that ferricyanide can easily

pass the outer membrane by its hydrophilicity (lines 177-178). Does that mean hydrophilic molecules can easier pass this membrane than hydrophobic molecules? Since LPS is an amphiphilic molecule consisting of both hydrophilic and hydrophobic regions it acts as a barrier for both hydrophilic and hydrophobic molecules. As far as I know there exists ways for both types of molecules to pass the outer membrane (Benz and Bauer, 1988, *Eur J Biochem* 176, 1-19; Wiener and Horanyi, 2011, *Proc Natl Acad Sci U S A*, 108, 10929-10930) and some hydrophobic drugs can diffuse across the lipid bilayer of the outer membrane (Delcour, 2009, *Biochim Biophys Acta*, 1794, 808-816). In lines 271-273 you wrote that an inhibition of the Calvin cycle also leads to a halt of oxygen evolution. I wonder how does this accompany with the fact that pCMB treated cells generated NADPH as fast as untreated cells. Inhibition of the water splitting will stop the non-cyclic electron transport and the production of NADPH. It cannot be explained either by oxidation of glycogen reserves in the absence of oxygenic photosynthesis since you wrote the inhibition of the Calvin cycle will prevent the generation of organic compounds and as a consequence no NADPH can be gained anymore by the pentose phosphate pathway of organic molecules (lines 275-278). Last I would like to mention that in several figures the lines were described to be purple, however, in my opinion their color is closer to violet. I think this paper should be published soon as it will help a lot the cyanobacterial community especially for increased released for biofuels as you mentioned in the end of the discussion. I wish you good luck. Many thanks once more, I really enjoyed it.

Reviewer #3:

Remarks to the Author:

The manuscript by Kusama et al. reports the increased photoelectrochemical output and reducing ability of a new mutant strain of *Synechocystis* sp. PCC 6803 that is lacking the outer membrane. The mutant is new, and its development is being reviewed in a different journal.

The idea behind the study is an interesting one, and the finding potentially quite important - it reaffirms that redox molecules are involved in cyanobacterial exoelectrogenesis and are shuttling electrons from the PETC to the membrane. The authors conducts a series of experiments to show that the reducing power of these cells lacking the outer membrane are greater; however, it would be more valuable if the authors could show how the electrons are passed from inside to the outside of the membrane and give more details about, for example- is it via passive or active diffusion? Is it the same redox-active compound? Is the mediation process compartmentalised? As it stands, it seems a bit obvious that removing the outer membrane (an obstacle between the PETC machineries and the electrode) would increase photocurrents..isolated components typically give higher photocurrents (e.g. isolated thylakoids gives rise to higher photocurrent densities than cells).

Overall, although the finding of the study is interesting for the field, manuscript is not ready for publication. It currently reads like an early draft with very poor-quality figures and many technical issues that require more consideration/experiments to validate the conclusions drawn. These include:

- 1) Characterisation of the cells, the cell-electrode interface and the number of cells participating in the photocurrent generation is lacking. To be more specific:
 - a. The difference in photocurrent output between the strains of cells must also be affected by differences in cell-electrode interactions. It is therefore important that characterisations of the cell-electrode interface been given (how are the cells sticking to the electrodes)? There is currently no images or other characterisation of the interface in this manuscript.
 - b. More information/characterisation regarding the mutants must be given for this paper. Is the outer membrane completely missing or loosely bound in some way? Do the mutants grow the same way/have the same amount of Chl a/have same dimensions as the wildtype? Does removing the outer membrane disturb the functionality of the periplasmic membrane? More importantly, are they as stable/robust as the wildtype? One of the main advantages of using whole cells is their robustness. We are given extremely little information about these mutants.

c. The normalisation of the photocurrent to the chlorophyll content (to infer the number of cells sticking to the electrode giving rise to the photocurrent) is missing. This is important for understanding if the same number of cells are contributing to the differences in output. The 35-fold enhancement may be partially due to more cells being able to closely contact the electrodes.

2) Line 81: 'Slr0688i generated 35 times as much photocurrent as dCas9 on average (Fig. 2(a)), and approx. 100 times as much photocurrent ($0.4 \mu\text{A}/\text{cm}^2$) as dCas9 ($0.007 \mu\text{A}/\text{cm}^2$) at maximum (Fig. 2(b)) with flat ITO electrodes.' How are the differences in photocurrent output calculated? Typically, the fairest section for photocurrent comparison is at the steady-state region (after the initial peaks). Could a comparison at this region be given instead?

3) The inhibitors results are poorly explained and give rise to confusing conclusions. For example, the results from the use of pCMB, a known inhibitor of the Calvin cycle, gives very strange results that are not satisfactorily explained. It was observed to enhance photocurrents by 30 folds, yet kills PSII activity, results in the same NADPH generation rate. From this, the authors explained that inhibiting the Calvin Cycle also inhibits PSII activity. However, why is it also enhancing photocurrents (where are so many electrons coming from if not from water oxidation?) and how is this showing that the electrons originate from downstream of NADPH? Does PMA, which also inhibits downstream of PSI also give similar enhancements or inhibit PSII? The observations of action of the inhibitors are highly confusing and the explanations are somewhat unsatisfying. I suggest that the authors either explain this much better or exclude it from the paper and publish it when a proper explanation can be found (since they do say that work on this is ongoing). I do not think the use of inhibitors add much value to the conclusions of the paper.

4) Line 163: 'Because the catalytic activity of a flat ITO electrode is very poor,' – please explain what does the catalytic activity of ITO mean in this context? ITO shouldn't be performing any catalysis in this system.

5) The ferricyanide assay used relies on the decrease in absorption of the reduced ferrocyanide species to infer electron transfer rates. However, this assay is prone to mis-interpretation since the decrease in ferricyanide absorption could also correspond to consumption/sequestration of the iron complex. Can the authors explain how they account for the consumption/sequestration affects, especially since the cells are now more readily penetrable by the ferricyanide complex.

6) Errors:

- a) In some figures, SE is used, and others SD. Could the authors justify the different analysis used?
- b) In some figures (Fig 4 and 6), no errors or replicates are mentioned. Could these be provided?
- c) In Fig2 a., the errors seem very large even with a large n (and using SE analysis). Could the authors comment on the reproducibility of these results?

In the revised manuscript, we have highlighted the revised text in yellow. Text deleted from the original version of the manuscript is highlighted in gray. Below we provide our point-by-point responses to each of the reviewers' comments.

Reviewer #1 (Remarks to the Author):

Reviewing the manuscript by Kusama et al 201C; A 35 fold 2026;..201D; submitted to Nature comm.

The authors used a formally generated mutant of Synechocystis, slr0688i, in which the cell membrane is deprived for the entrance and moving out of metabolites and proteins. Using this mutant in a biophotovoltaic cell generated 35 fold enhancement in the EET current. The study of the EET phenomenon in bacteria, cyanobacteria and algae is intensive and important in order to be able to generate an electricity producing photo-cell that will make use of live microorganisms and light to generate green energy. However, several points prevent this manuscript of being a novel one and providing new addition to the field.

1. The currents produced in the photoelectric cell are significantly lower than the current produced by cyanobacteria in other publications. This is true even to the enhanced currents generated by the mutant in this work. This is probably since an ITO anode was used. It is already well established that the carbon-graphite electrode generates more than x10 higher currents.

<A1-1>

We appreciate that there are two approaches to improve the cyanobacterial extracellular electron transfer (EET) current, i) manipulating the cell itself to improve its electrogenic activity, and ii) optimizing electrodes and/or the cell–electrode interaction, as Reviewer I indicated. In this study we focused on the former issue and our goal was to define the effects of outer membrane deprivation on the EET activity. As such, we carefully evaluated the photocurrent generation, ferricyanide reduction, and electron donation capacity to other bacteria, but did not conduct optimization of the electrodes. The results clearly indicated that outer membrane deprivation is a promising technology to improve the efficiency of BPV systems.

As discussed in the original version of the manuscript (lines 353–361), the photocurrent is expected to be further enhanced by combining various other approaches such as optimization and/or modification of electrodes, strengthening the cell–electrode interaction, or optimizing the cell culture condition. However, these considerations were beyond the scope of the present study and are appropriate topics of future research. To clarify the aim and scope of the present study, we revised the text on lines 82–85 and 341–348 of the revised manuscript.

2. It was already shown that manipulating the cell membrane by physically means like an osmotic treatment or low-pressure microfluidaizer treatment significantly enhances the EET in Synechocystis (ref 15). Therefore, this study repeats this phenomenon in a genetic eng manipulated technique.

<A1-2>

We perceive two main differences, which we believe address the importance of our study, between physically treated cells (Saper et al., 2018; reference 15) and genetically manipulated cells deprived of an outer membrane (our present study) as follows. I) The effect of physical treatment is assumed not to be long-lasting but temporal, because living cells are intrinsically self-repairing (e.g., the case of membrane-damaged *Escherichia coli*; Chilton et al., 2001, International Journal of Food Microbiology, 71: 101–104). In contrast, genetically manipulated cells can proliferate while the effect of outer membrane deprivation is maintained. This difference is potentially decisive when considering the robustness of biophotovoltaics, especially from the point of view of biotechnological applications. II) The intracellular electron flux contributing to EET seems to differ because the EET of physically treated cells is enhanced by the presence of DCMU [Saper et al. (2018) speculated that this effect was caused either by accelerating the forward electron flow of respiration or enhancing the release of a mediator]. If our present study replicated this phenomenon, the DCMU-enhanced photocurrent would have been observed in our experiments. However, the effect was the opposite: DCMU abolished the photocurrent (Fig. 2(a), p. 60). This intriguing difference implies that the cellular electron flow may be affected by physiological conditions and perhaps by the cell surface status, thus providing additional topics for future research. Accordingly, we consider that our present study provides novel advances in the EET research field. To emphasize the above-mentioned difference, we modified the text on lines 362–363 in the revised manuscript.

3. It has been recently reported that the electron mediator in this organism is NAPH (ref 17). Therefore, the following work should study this component as the electricity generated component.

<A1-3>

We performed additional experiments to investigate the site from where electrons stem, and accordingly have made major revision of the manuscript (lines 135–243). We concluded that electrons stem from the pathway(s) downstream of PSI. Currently, NADPH appears not to serve as a mediator under our experimental conditions because i) reactivity between ITO and NADPH is low [the study by Shlosberg et al. (2021) (reference 17) used a graphite electrode], ii) our preceding

metabolomic study of the culture supernatant of slr0688i cells failed to detect NADPH (Kojima and Okumura, 2020; reference 25), and iii) NADPH fluorescence intensity was not significantly affected in pCMB-treated cells despite that they showed greatly enhanced photocurrent generation. Furthermore, as we mentioned in our response A1-2 to the comment from Reviewer 1), the intracellular electron flux leading to EET may differ because the photocurrent demonstrated by Shlosberg et al. (2021) was enhanced in the presence of DCMU, whereas the photocurrent was decreased by DCMU in our study (Fig. 2(a), p. 60). We therefore revised the text on lines 407–423 of the revised manuscript.

Minor notes:

1. The nature of the membrane deprived mutant sl0688i is not describe and the reader should go to a manuscript in bioArxiv (ref 25) to study the details. It should be described in the present manuscript in details and the molecular mechanism discussed in the discussion. Perhaps it would be better and naturally to combine the two manuscripts.

<A1-4>

According to the reviewer's comment, we added a description of the slr0688i mutant to improve the readability (lines 363–372). Our preceding study (Kojima and Okumura, 2020; reference 25) focused solely on physiological aspects of the membrane-deprived mutant, whereas the present study exclusively focused on EET activity of this mutant. An additional experiment (the hydrophobicity assay described on lines 99–107) reaffirmed that the cell surface properties of the slr0688i mutant were drastically changed from that of dCas9 cells. Because the objectives of these two studies differ, we consider that it is not unreasonable to describe the findings in separate manuscripts. This reasoning was also supported by Reviewer 2, with the following comment: "I am clearly impressed first about the deprivation of a cyanobacterium's outer membrane for the first time by your group in a separate work and second the increased external electron transfer by the same mutant strain discovered in this work."

2. Line 41. What is “higher photosynthetic activity than plants”. In what sense?

<A1-5>

In response to the reviewer's comment, we modified the wording of the phrase to "higher energy conversion efficiency of photosynthesis" to clarify the intended meaning (lines 42–43). As demonstrated in references 5–7 (Huntley and Redalje, 2007; Li et al., 2008; Dismukes et al., 2008),

the conversion efficiency from solar energy to biomass is reported to be higher in microalgae than that in terrestrial plants.

3. Figure 1. This is a nice schematic drawing but add no result to the work. Could be displayed as a panel in another figure.

<A1-6>

In accordance with this comment, we merged Figs. 1 and 2 from the original version of the manuscript as Fig. 1 in the revised manuscript (p. 56).

4. Figure 2. As said above, the phot-currents are significantly low compared to what already achieved with live *Synechocystis*.

<A1-7>

As explained in our response A1-1 to Reviewer 1's comment, the main purpose of the present study was to determine the effects of outer membrane deprivation on EET activity. Therefore, we simply used cell suspensions and performed the photocurrent measurements by placing the cell suspension by gravity onto an unmodified electrode. The photocurrent *per se* could be further enhanced by optimizing the electrodes and the cell–electrode interaction in future research, as we explained already in the original version of the manuscript (lines 353–361 of the revised manuscript) and mention in our response A1-1. To clarify and emphasize this point, we revised the text on lines 82–85 and 341–348 in the revised manuscript.

5. Figure 3. DCMU has been shown in other work to increase the photo-current generated by live *Synechocystis*. Discuss what is the possible difference.

<A1-8>

We are aware that there are previous studies that showed DCMU enhances the photocurrent (Saper et al., 2018; reference 15; Shlosberg et al., 2021; reference 17), but also studies that reported DCMU inhibits the photocurrent of *Synechosystis* sp. PCC 6803 (Bombelli et al., 2011; reference 11; Cereda et al., 2014; reference 13; Zhang et al., 2018; reference 14). Hence, DCMU has not been consistently reported to enhance the photocurrent. The action of DCMU is intriguing and may depend on the physiological condition of the cells. Multiple growth condition parameters can affect the cellular physiology and intracellular electron flow; for example, light intensity is known to significantly affect the amount of PSI (Kopečná et al., 2012, *Plant Physiology*, 160: 2239–2250). The shape of the culture vessels and the speed of shaking influence the aeration efficiency and thus affect the CO₂ availability,

which is one of the decisive factors in cyanobacterial physiology. The cell–electrode interaction, and perhaps the cell surface properties, may also influence the intracellular electron flow. Saper et al. (2018) (reference 15) suggested the currently unknown action(s) of DCMU may involve accelerating the forward electron flow of respiration or enhancing the electron mediator release. This complicated situation poses difficulties in presenting a clear discussion. We consider that the action of DCMU remains to be elucidated and thus we refrained from providing a speculative discussion in the manuscript. We have explained the current uncertainty regarding the EET mechanism and DCMU action in the revised manuscript (lines 407–423).

Reviewer #2 (Remarks to the Author):

Dear researchers of that paper!

First I want to tell you that I was very excited by the topic. Since researchers always have to prove the impact of their research onto the society, I think this paper will help to even further increase the economic importance of cyanobacteria (after approaches to use them for the gain of biofuel within the last decade). As my topic in the past has been heterotrophy and transport across the cytoplasmic membrane, I am clearly impressed first about the deprivation of a cyanobacterium's outer membrane for the first time by your group in a separate work and second the increased external electron transfer by the same mutant strain discovered in this work. So over all there is hardly anything I can criticize.

However, I would be interested, how long it takes after resuspension in fresh BG11 medium to establish photocurrent again after enough redox active compounds have been secreted, since your cells have also been inoculated in fresh medium at the beginning of cultivation before the actual experiment.

<A2-1>

In this study, cells were diluted to $OD_{730} = 0.1$ at the beginning of the culture period and 4-day cultures were used for photocurrent measurements (3 to 4-day cultures were used for the other measurements). We did not observe photocurrent generation by younger cultures. In accordance with the comment, we clarified this information in the Materials and Methods (lines 487–488, 549, 664).

You mentioned that KCN inhibits all respiratory terminal oxidases but also plastocyanine (lines 148-149). However, cytochrome c can also serve as an electron carrier in photosynthesis thereby completely replacing plastocyanine in its function since plastocyanine deletions have already been performed in *Synechocystis* sp. PCC 6803 (Pils and Schmetterer, 2001, FEMS Microbiol Lett 203, 217-222). In Fig. 3b cytochrome c should also be displayed.

<A2-2>

It is known that plastocyanin is a copper concentration-dependent protein (Zhang et al., 1992; reference 81). The BG11 medium used in the present study contains 0.3 μM copper, under which plastocyanin serves as a dominant electron carrier (shown in Fig. 5 of Zhang et al., 1992). Following the reviewer's comment, we added cytochrome *c* in Fig. 2(b) (on p. 60) and revised the Fig. 2 legend on lines 1006–1008.

You write that ferricyanide can easily pass the outer membrane by its hydrophilicity (lines 177-178). Does that mean hydrophilic molecules can easier pass this membrane than hydrophobic molecules? Since LPS is an amphiphilic molecule consisting of both hydrophilic and hydrophobic regions it acts as a barrier for both hydrophilic and hydrophobic molecules. As far as I know there exists ways for both types of molecules to pass the outer membrane (Benz and Bauer,1988, Eur J Biochem 176, 1-19; Wiener and Horanyi, 2011, Proc Natl Acad Sci U S A, 108, 10929-10930) and some hydrophobic drugs can diffuse across the lipid bilayer of the outer membrane (Delcour, 2009, Biochim Biophys Acta, 1794, 808-816).

<A2-3>

We apologize for providing confusing sentences in the original manuscript. We removed the word “hydrophilicity” and corrected the text on lines 265–272. Our intended meaning was that it has long been thought that ferricyanide freely passes through the outer membrane (for example, as mentioned by McCormick et al., 2015; reference 2). However, our present study revealed that the outer membrane serves as a barrier to ferricyanide access to the cytoplasmic membrane. This result shows good agreement with the previous study by Kowata et al. (2017) (reference 20), who showed that the cyanobacterial outer membrane serves as a low-permeability molecular sieve that prevents the passage of hydrophilic organic molecules.

In lines 271-273 you wrote that an inhibition of the Calvin cycle also leads to a halt of oxygen evolution. I wonder how does this accompany with the fact that pCMB treated cells generated NADPH as fast as untreated cells. Inhibition of the water splitting will stop the non-cyclic electron transport and the production of NADPH. It cannot be explained either by oxidation of glycogen reserves in the absence of oxygenic photosynthesis since you wrote the inhibition of the Calvin cycle will prevent the generation of organic compounds and as a consequence no NADPH can be gained anymore by the pentose phosphate pathway of organic molecules (lines 275-278).

<A2-4>

We performed additional experiments to determine the effects of pCMB and accordingly made major revisions to the manuscript (lines 135–243). By measuring NADPH fluorescence, we confirmed that pCMB does not inhibit NADPH generation at the onset of illumination (Supplementary Fig. 9). However, prolonged light illumination (>1 min) of pCMB-treated cells caused a decrease in oxygen-evolving activity and effective quantum yield of PSII [Y(II)] (Supplementary Figs. 6 and 7). These are common phenomena in cells in which the Calvin cycle is inhibited (Takahashi and Murata, 2005; reference 41). Consistent with this, we noted that the pCMB-enhanced photocurrent was most pronounced shortly after the start of illumination, and the photocurrent gradually declined close to the

initial background level. We consider that pCMB perturbs the consumption of photosynthetically generated reducing energy presumably by inhibiting the Calvin cycle, and this temporarily reinforces the reduction of possible electron mediator(s) as the alternative consumption route, thereby enhancing the photocurrent. To elaborate on the above point, we revised the text on lines 155–169 in the revised manuscript.

Last I would like to mention that in several figures the lines were described to be purple, however, in my opinion their color is closer to violet.

I think this paper should be published soon as it will help a lot the cyanobacterial community especially for increased released for biofuels as you mentioned in the end of the discussion. I wish you good luck. Many thanks once more, I really enjoyed it.

<A2-5>

In accordance with the reviewer's comment, we replaced "purple" with "violet" in the revised manuscript as follows: line 999 in the manuscript text; lines 65, 103, and 206 of the Supplementary Information.

Reviewer #3 (Remarks to the Author):

The manuscript by Kusama et al. reports the increased photoelectrochemical output and reducing ability of a new mutant strain of *Synechocystis* sp. PCC 6803 that is lacking the outer membrane. The mutant is new, and its development is being reviewed in a different journal.

The idea behind the study is an interesting one, and the finding potentially quite important - it re-affirms that redox molecules are involved in cyanobacterial exoelectrogenesis and are shuttling electrons from the PETC to the membrane. The authors conducts a series of experiments to show that the reducing power of these cells lacking the outer membrane are greater; however, it would be more valuable if the authors could show how the electrons are passed from inside to the outside of the membrane and give more details about, for example- is it via passive or active diffusion? Is it the same redox-active compound? Is the mediation process compartmentalised? As it stands, it seems a bit obvious that removing the outer membrane (an obstacle between the PETC machineries and the electrode) would increase photocurrents…isolated components typically give higher photocurrents (e.g. isolated thylakoids gives rise to higher photocurrent densities than cells). Overall, although the finding of the study is interesting for the field, manuscript is not ready for publication.

It currently reads like an early draft with very poor-quality figures and many technical issues that require more consideration/experiments to validate the conclusions drawn. These include: 1) Characterisation of the cells, the cell-electrode interface and the number of cells participating in the photocurrent generation is lacking. To be more specific:

The difference in photocurrent output between the strains of cells must also be affected by differences in cell-electrode interactions. It is therefore important that characterisations of the cell-electrode interface been given (how are the cells sticking to the electrodes)? There is currently no images or other characterisation of the interface in this manuscript.

<A3-1>

Regarding the reviewer's overall comment on the electron mediation process, throughout our study we used cell suspensions for electrochemical measurements and did not employ any procedure to make the cells adhere to the electrodes. When the culture supernatant was removed and the cells were resuspended in fresh BG11 medium, the ability of photocurrent generation was lost (Fig. 1(d)). This observation indicates that the electron donation to the electrode requires the presence of the mediator(s) in the culture supernatant. Thus, photocurrent generation consists of two steps: the intracellular electrogenesis and the extracellular, mediated electron transfer. McCormick et al. (2015) (reference 2) also proposed this type of EET, where some cytoplasmic membrane protein(s) accepts

intracellular electrons and donates them to different extracellular mediators. We agree that identifying the mediator(s) is desirable and this will provide an insight on how the mediator(s) moves between cells and an external space, but we believe that this should be the topic of future research; the objective of the present study was solely focused on the effect of outer membrane deprivation on EET activity.

Regarding the reviewer's comment 1), our study was focused on the EET activity of the cells but not on the cell–electrode interaction (as we mentioned in our response A1-1 to Reviewer 1's comment). Therefore, we did not employ a procedure to make the cells adhere to the electrode surface nor do we have images of cells adhering to the electrodes. We used cell suspensions and took great care in employing the same amount of cells for measurements by adjusting the OD₇₃₀ (4 mL cell suspensions of OD₇₃₀ = 1.5 were used, corresponding to approximately 6.0×10^8 cells (33.6 µg Chl) and 2.4×10^9 cells (27.6 µg Chl) for slr0688i and dCas9, respectively). The use of cell suspension-based measurement system inevitably led to an unstable and variable interaction between mediator(s) reduced at the cell surface and an electrode. To explain this, we revised the text on lines 82–85, 341–348, 488 and 493–498 in the revised manuscript.

As the hydrophobicity of the cell surface is one of the decisive factors for cell–electrode interaction, we additionally performed a hydrophobicity assay by mixing slr0688i or dCas9 cell cultures with *p*-xylene. The results revealed that $49.0\% \pm 8.7\%$ of the cells adsorbed to *p*-xylene in the case of dCas9, whereas only $6.3\% \pm 1.8\%$ of the cells were adsorbed in the case of slr0688i. The finding that the hydrophobicity of slr0688i is significantly lower than that of dCas9, and the fact that hydrophobic cells are more adhesive to the ITO substrate (Bayouhd et al., 2006; reference 27; Ozkan and Berberoglu, 2013; reference 28) suggest that cell–electrode interaction is not enhanced in the slr0688i mutant. Therefore, the observed 35-fold enhancement in current generation is attributable to outer membrane deprivation, not to enhanced interaction between cells and an electrode. To elaborate on this point, we revised the text on lines 99–109, added the procedure for the hydrophobicity assay in the Materials and Methods section on lines 539–547, and added a new Supplementary Fig. 2 to p. 2 of the Supplementary Information.

Finally, with regard to the reviewer's comment on the figure quality, we tried to improve the quality of the following figures in the revised manuscript: Figs. 1, 2, 3, and 4; Supplementary Figs. 1, 4, 5, 7, 10, 13, 14, and 18.

b. More information/characterisation regarding the mutants must be given for this paper. Is the outer membrane completely missing or loosely bound in some way? Do the mutants grow the same way/have the same amount of Chl a/have same dimensions as the wildtype? Does removing the outer membrane disturb the functionality of the periplasmic membrane? More importantly, are they as stable/robust as the wildtype? One of the main advantages of using whole cells is their robustness. We are given extremely little information about these mutants.

<A3-2>

In accordance with the reviewer's comment, we added information on the outer membrane-deprived mutants to the revised manuscript (lines 363–372). As judged from electron-micrographs of ultrathin-sectioned slr0688i cells, the outer membrane appears to be missing from almost 70% of the cell surface. On the remainder of the cell surface, the outer membrane is loosely associated with the cell (Kojima and Okumura, 2020; reference 25). The mutant cells can photoautotrophically proliferate, but the growth rate is retarded compared with that of the wild type. The photosynthesis activity is comparable to that of the wild type (this was also shown in the present study in Supplementary Fig. 1). The cytoplasmic membrane is functional at least for sustaining cell growth. It should be noted that outer membrane deprivation liberates the periplasmic proteins into the external environment; this may partially interfere with the function of the cytoplasmic membrane because various cytoplasmic membrane proteins interact with the periplasmic proteins.

The chlorophyll contents of the two cell strains were $5.6 \pm 0.3 \mu\text{g Chl (mL OD}_{730})^{-1}$ in slr0688i and $4.6 \pm 0.2 \mu\text{g Chl (mL OD}_{730})^{-1}$ in dCas9. We revised the text on lines 488 and 493–498 to include these data in the revised manuscript.

c. The normalisation of the photocurrent to the chlorophyll content (to infer the number of cells sticking to the electrode giving rise to the photocurrent) is missing. This is important for understanding if the same number of cells are contributing to the differences in output. The 35-fold enhancement may be partially due to more cells being able to closely contact the electrodes.

<A3-3>

We carefully adjusted 4 mL of cell suspensions to $\text{OD}_{730} = 1.5$ (corresponding to approximately 6.0×10^8 cells (33.6 $\mu\text{g Chl}$) and 2.4×10^9 cells (27.6 $\mu\text{g Chl}$) for slr0688i and dCas9, respectively) to employ the same amount of cells for all photocurrent measurements. Because our present study was focused solely on the EET activity of the cells and not on the cell–electrode interaction (as we mentioned also in our response A1-1 to Reviewer 1's comment), we did not employ a procedure to make the cells adhere to the electrode surface. Moreover, as mentioned in our response A3-1 to Reviewer 3's comment, the slr0688i cell surface was less hydrophobic than that of dCas9 cells, suggesting that cell–electrode interactions are not enhanced. Thus, the 35-fold enhancement in current generation is attributable to outer membrane deprivation. As described in our responses A1-1 and A3-1, we modified the text on lines 82–85, 99–109, 341–348, 488 and 493–498 of the revised manuscript to explain these points.

2) Line 81: $\mu\text{A/cm}^2$; Slr0688i generated 35 times as much photocurrent as dCas9 on average (Fig. 2(a)), and approx. 100 times as much photocurrent (0.4 $\mu\text{A/cm}^2$) as dCas9 (0.007

μA/cm²) at maximum (Fig. 2(b)) with flat ITO electrodes.’ How are the differences in photocurrent output calculated? Typically, the fairest section for photocurrent comparison is at the steady-state region (after the initial peaks). Could a comparison at this region be given instead?

<A3-4>

We observed that the photocurrent produced by the outer membrane-deprived cells did not necessarily show the initial peaks (as illustrated by Fig. 1(b) and Supplementary Fig. 5). The current was most pronounced in the initial period of illumination (typically during approximately 40 s after the onset of illumination), and then gradually and unstably decreased during prolonged illumination. Thus, we chose the current level obtained during the time period of 0 to 70 s in Fig. 1(b) for the comparison. The calculation yielding 35-fold enhancement in the photocurrent was conducted at the time point of 64 s. To clarify this, we revised the text on lines 85–89 of the revised manuscript. In addition, we removed the description regarding the calculation yielding 100-fold enhancement using representative data so as to make the comparison fairer, consistent with the reviewer’s comment.

3) The inhibitors results are poorly explained and give rise to confusing conclusions. For example, the results from the use of pCMB, a known inhibitor of the Calvin cycle, gives very strange results that are not satisfactorily explained. It was observed to enhance photocurrents by 30 folds, yet kills PSII activity, results in the same NADPH generation rate. From this, the authors explained that inhibiting the Calvin Cycle also inhibits PSII activity. However, why is it also enhancing photocurrents (where are so many electrons coming from if not from water oxidation?) and how is this showing that the electrons originate from downstream of NADPH? Does PMA, which also inhibits downstream of PSI also give similar enhancements or inhibit PSII? The observations of action of the inhibitors are highly confusing and the explanations are somewhat unsatisfying. I suggest that the authors either explain this much better or exclude it from the paper and publish it when a proper explanation can be found (since they do say that work on this is ongoing). I do not think the use of inhibitors add much value to the conclusions of the paper.

<A3-5>

We performed additional experiments to determine the effects of pCMB as well as other inhibitors. Accordingly, major revision of the manuscript has been done (lines 135–243 and 373–407). The findings in the revised manuscript now suggest that electrons stem from a pathway(s) downstream of PSI. By measuring NADPH fluorescence and P700 redox kinetics, we confirmed that pCMB does not inhibit NADPH generation during the onset of short-period light illumination (Supplementary Fig. 9). However, prolonged light illumination (>1 min) of pCMB-treated cells caused a decrease in oxygen-evolving activity and effective quantum yield of PSII [Y(II)] (Supplementary Figs. 6 and 7). These are

common phenomena in cells in which the Calvin cycle is inhibited (Takahashi and Murata, 2005; reference 41). Consistent with this, we noted that the pCMB-enhanced photocurrent was most pronounced shortly after the start of illumination, and subsequently the photocurrent gradually declined close to the initial background level (Fig. 2(a)), reflecting the inhibition of photosynthetic electron generation. We confirmed that this current trace pattern was reproducible (Supplementary Fig. 5). We consider that pCMB perturbs the consumption of photosynthetically generated reducing energy presumably by inhibiting the Calvin cycle, and this temporarily reinforces the reduction of possible electron mediator(s) as the alternative consumption route, thereby enhancing the photocurrent. Detailed description and discussion of these results are included in the revised manuscript (lines 135–201 and 373–402). In addition, we revised the text on line 25 (abstract) and the following figures have been modified: Fig. 2 and Supplementary Figs. 5–9.

The Materials and Methods section has been revised to describe the procedures used for these additional experiments (lines 533–538 and 565–633).

The additional experiments indicated that PMA inhibits electron flow via PSII under prolonged illumination, as does pCMB (Supplementary Figs. 6 and 7) but the current is eliminated both in the dark and under illumination (Fig. 2(a) and Supplementary Fig. 5). NADPH fluorescence measurements (Supplementary Fig. 9(b)) confirmed that the inhibitory site of PMA is Fd/FNR as reported previously (Honeycutt and Krogmann, 1972; reference 30; Torimura et al., 1992; reference 39). Revisions to the text to describe these results are included on lines 178–183 of the revised manuscript.

4) Line 163: Because the catalytic activity of a flat ITO electrode is very poor, please explain what does the catalytic activity of ITO mean in this context? ITO shouldn't be performing any catalysis in this system.

<A3-6>

We apologize for the confusing sentence in the original manuscript. The purpose of the ferricyanide reduction assay was to evaluate EET activity that could not be fully detected by photocurrent measurements using unmodified ITO electrodes. Regarding this aim, using an easily reducible, artificial electron mediator (ferricyanide) can provide more accurate information on EET activity than just placing the cell suspension onto the ITO electrode; the latter approach could be affected by multiple factors, such as cell- and/or mediator-electrode interaction, as commented by Reviewer 3. To elaborate on this point, we revised the text on lines 246–254 in the revised manuscript.

5) The ferricyanide assay used relies on the decrease in absorption of the reduced ferrocyanide species to infer electron transfer rates. However, this assay is prone to mis-interpretation since

the decrease in ferricyanide absorption could also correspond to consumption/sequestration of the iron complex. Can the authors explain how they account for the consumption/sequestration affects, especially since the cells are now more readily penetrable by the ferricyanide complex.

<A3-7>

In accordance with the reviewer's comment, we performed additional experiments to quantify the total Fe concentration using inductively coupled plasma mass spectrometry (ICP-MS) as described below. The results confirmed that consumption/sequestration of the iron complex did not occur during the assay. In the additional experiment, *Synechocystis* cell suspensions adjusted to $OD_{730} = 1.0$ were mixed with 1 mM potassium ferricyanide, followed by incubation at 30 °C with constant shaking at 140 rpm under illumination ($50 \mu\text{mol photons m}^{-2} \text{s}^{-1}$). Here, the cells were washed and resuspended with iron-free BG11 medium. Before ($t = 0$) and after ($t = 2$ h) incubation, the concentrations of ferricyanide and Fe in filtrated supernatants were measured with a spectrophotometer and ICP-MS, respectively. The results revealed that the concentration of Fe was not changed during the assay. In the revised manuscript, we added a new Supplementary Fig. 11 to present the results of the above-mentioned experiment, and we revised the text on lines 272–278, 556–564, and 931–932.

6) Errors:

a) In some figures, SE is used, and others SD. Could the authors justify the different analysis used?

<A3-8>

We used the SD in Fig. 3 because here, both variability and average values were of interest to us. On the other hand, we used the SE in the photocurrent comparison (Fig. 1(b)). The rate of electron donation from reduced mediator(s) to an electrode is affected strongly by interaction between them. In the present study, we used cell suspensions placed by gravity onto the electrode for photocurrent measurement to focus exclusively on the EET activity of the cells but not on the cell–electrode interaction; this cell suspension-based measurement inevitably leads to an unstable interaction between mediator(s) reduced by the cell surface and an electrode. In addition, numerous growth condition parameters (including a slight difference in the distance between a light source and culture vessels) and the physiological condition of the cells (which is even affected by the condition of precultured cells used for inoculation) affect the amount of co-existing metabolites other than mediator(s) in the suspension, the extent of outer membrane deprivation, or how the remaining outer membrane associates with the cell surface; these factors also affect the interaction between mediators and an electrode. Consequently, although we made extensive efforts to stabilize the cell culture condition, photocurrent amplitudes still varied. Given that the priority of the measurements was to

compare the averaged levels of photocurrent generation between the outer membrane-deprived cells and the wild type, we used the SE to display the accuracy of the calculated means. Please note that, to our knowledge, no previous studies have reported the photocurrent trace of *Synechocystis* sp. PCC 6803 with error bars for repetitive measurements. Thus, we cannot know whether the SE values for our photocurrent measurements are larger than those of other studies. Because it is more appropriate to show the average values $\pm 2SE$ instead of SE, Fig. 1(b) was modified to present means $\pm 2SE$ in the revised manuscript.

b) In some figures (Fig 4 and 6), no errors or replicates are mentioned. Could these be provided?

<A3-9>

We have included a new Supplementary Fig. 12 to show other biological replicates of the experiment shown in Fig. 4(a) (which was Fig. 6(a) in the original version of the manuscript).

We have included a new Supplementary Fig. 15 to show other biological replicates of the experiment shown in Fig. 4(b) (which was Fig. 6(b) of the original version of the manuscript).

Although we performed the experiment described in Fig. 4 of the original version of the manuscript with two biological replicates, following the major revision to the manuscript, this figure is not included in the revised manuscript.

c) In Fig2 a., the errors seem very large even with a large n (and using SE analysis). Could the authors comment on the reproducibility of these results?

<A3-10>

Given that the current amplitude, i.e., the rate of electron donation from reduced mediator(s) to an electrode, is affected strongly by the interaction between them, there are two possible reasons for the large errors. I) We used cell suspensions placed by gravity onto the electrode for photocurrent measurement to focus exclusively on the EET activity of the cells but not on the cell–electrode interaction. This cell suspension-based measurement system inevitably led to an unstable and variable interaction between mediator(s) reduced at the cell surface and an electrode. II) As we mentioned in our response A3-8, numerous growth condition parameters as well as the physiological condition of the cells affect the amount of co-existing metabolites other than mediator(s) in the suspension, the extent of the outer membrane deprivation, or the association of the remaining outer membrane with the cell surface; these factors also affect interaction between mediators and an electrode. It is worth noting that, to our knowledge, no other studies have reported the amperometric *i-t* curve of *Synechocystis* sp. PCC 6803 cell suspensions with error bars for repetitive

measurements, possibly because of their low electrogenic activity, i.e., small current. Thus, we do not know whether the SE values of our photocurrent measurements are larger than those of previous studies. As mentioned in our response A3-8 to Reviewer 3's comment, because it is more appropriate to show the average values $\pm 2SE$ instead of the SE, Fig. 1(b) was modified to show the means $\pm 2SE$ in the revised manuscript.

Reviewers' Comments:

Reviewer #1:

Remarks to the Author:

The authors provided explanations and answers to most of the questions and issues that were raised in the first reviewing cycle.

A major one was that replacing the ITO with a carbon paper anode increased the current to that usually reported in the literature for these systems. With this nice current, the question is whether or not the experiments performed with the ITO, resulting with significantly very low and almost neglectable currents, actually represent the same phenomena as if the experiments would have been performed with carbon based anode? The way to answer this important question would be to perform the experiments with a BPT containing a carbon type anode and measuring a substantial amount of current.

The description of the genetic engineered strain is still poor and not enough to gain the reader with the understanding of the differences.

Measuring NADPH using fluorescence as it was done here would be not sensitive enough to see the small differences generating the current. The enzymatic detection methods (sigma kit, for example) or the two-dimension fluorescent mapping method should be used.

Reviewer #2:

Remarks to the Author:

Dear Researchers!

Thank you very much for correcting most of my remarks. Now my doubts have been overwhelmingly dissolved. There are only some minor remarks I would like to perform.

Zhang et al. (1992) concluded that copper determines the ratio between plastocyanin and cytochrome c, however, 0.3 μM copper (the concentration of standard BG11 medium used in your paper) allows the expression of both carrier molecules, whereas no expression of cytochrome c was detected at 1 μM copper in the medium. Nevertheless Zhang et al. (1994, J Biol Chem 269, 5036-5042) stated that even 1 μM copper did not completely abolish the expression of cytochrome c, which explains that cytochrome c or plastocyanin single mutants can grow in conditions that strongly favor the expression of the deleted gene.

Besides I would like to suggest to modify Fig. 2 to include the splitting of water below PSII as you did the reversion above COX and Cyd. You wrote that inside the yellow dashed line there are the components where EET is inhibited by GA but not by pCMB, however, the Calvin cycle is included within this line although it is marked to be inhibited by pCMB. I also wonder why you mark the oxidative pentose phosphate pathway (glycolysis and TCA cycle) in this figure to be inhibited by GA but not by pCMB although you think that pCMB also inhibits the oxidative pentose phosphate pathway (lines 418 – 421). You write that the P700 oxidation level under illumination with far red light was elevated in GA-treated cells, however, this was even more true to PMA (supplementary figure 8). How does this fit with its proposed role to inhibit FNR. If no electrons can be transferred from Fd to NADP+ anymore, P700 should be in a more reduced state. And why does PMA prevent the consumption of NADPH in the dark (supplementary figure 8) if it does not inhibit NDH1 or NDH2 (no such mark in fig. 2).

Finally in line 156 of supplemental material you should add (green) after DCMU and split supplementary figure 8 into a, b and c as you refer to supplementary Fig. 8(a) in line 174 of your main manuscript. Besides it would be helpful to describe the oxidation state of the (OOH) in line 540 to allow the same total charge on both sides of the equation.

Overall I wish you once more good luck and hope you can publish it as soon as possible.

Reviewer #3:

Remarks to the Author:

The revisions made by Nakanishi et al. have addressed most of the concerns of the referees. The authors are to be commended on these efforts, some of which are above and beyond. I think this manuscript is ultimately deserving to be published in Nat. Comm. if the main results can be validated; however there remain important points that remain poorly addressed by the authors regarding the electrochemical approach, which leaves this reviewer unconvinced by the interpretation of the data. Specifically, these issues are:

1) The authors state: 'We used cell suspensions placed by gravity onto the electrode for photocurrent measurement to focus exclusively on the EET activity of the cells but not on the cell-electrode interaction'.

This reviewer is very concerned about the electrochemical approach and the interpretations of the results in Figure 1. If the authors want to 'focus exclusively on the EET activity of the cells', then the best practice would be to use a stirred cell suspension, perhaps with the addition of a non-cell permeable mediator to further aid the electron transfer from the solution bulk to the electrode. Only then would there be minimal cell-electrode interaction at play. However, in the current method (letting the cells settle on the electrodes over time with the aid of gravity), there will be inevitably a layer of cells interacting with the electrode with multi-layers above. It is the formation of this first layer of cells that contributes most to the photocurrent and the multilayers above will contribute minimally to the photocurrent. Therefore, very precise cell loadings to the device (as claimed by the authors) is not sufficient for providing information on how many cells and how the cell-electrode interface contribute to the photocurrent. This is problematic because the authors are making very precise quantitative conclusions (35-fold enhancement compared to wildtype etc), which is not appropriate using this qualitative approach for performing bioelectrochemistry. For the authors to attribute the photocurrent output differences to differences in the EET activity alone is therefore inappropriate (there is not enough information to support this).

Some suggestions for addressing this problem:

i) Unless the method can be revised (for example with properly immobilised cells), please rephrase the discussion of the results to be more transparent about the limitations of this cell-loading approach. I would suggest re-wording the title as well to remove the '35-fold' enhancement,
ii) Take some representative SEM or microscopy images of the cells (for both wt and mutant) covering the bottom of the device to show information including – how many cell multilayers are there? What is the total coverage of the electrode? Is it homogenous distribution or not?

2) The interpretation of the chronoamperometry photocurrents is highly unconvincing. As mentioned by reviewer 3, 'typically, the fairest section for photocurrent comparison is at the steady-state region (after the initial peaks)'. The authors did not pick this steady state region for analysis (which according to Fig 1 (c) is at the 500s time point), instead picking the transient peak at around 70s time point.

The authors' response to this is that 'We observed that the photocurrent produced by the outer membrane-deprived cells did not necessarily show the initial peaks. However, it is clear looking at Fig 1 (c) and (e) that there is an initial peak- it is at 70 s, which is what the authors picked for analysis and cautioned against by reviewer 3.

Even more problematic is the interpretation of the new carbon cloth data Fig 1 (e). It can be seen that after 'light on', the dark current does not return to baseline, but instead stays very high - very similar to the photocurrent. Unless the authors can show the dark current returning fully to the original baseline, they cannot claim that the 30 μAcm^{-2} photocurrent output that they put forward as the new benchmark output.

Overall, although the non-electrochemical data put forward in this manuscript convincingly shows that the mutant cell line has higher EET activity, the electrochemical data put forward is not sufficiently convincing to back the quantitative claims of the paper and instill confidence regarding the reproducibility of the outputs.

In the manuscript, we have highlighted the revised text in pale blue (to be distinguished from those already marked in yellow in NCOMMS-21-01109B). Text deleted either from the original version of the manuscript or NCOMMS-21-01109B is highlighted in gray. Below we provide our point-by-point responses to each of the reviewers' comments.

Reviewer #1 (Remarks to the Author):

The authors provided explanations and answers to most of the questions and issues that were raised in the first reviewing cycle.

A major one was that replacing the ITO with a carbon paper anode increased the current to that usually reported in the literature for these systems. With this nice current, the question is whether or not the experiments performed with the ITO, resulting with significantly very low and almost neglectable currents, actually represent the same phenomena as if the experiments would have been performed with carbon based anode? The way to answer this important question would be to perform the experiments with a BPT containing a carbon type anode and measuring a substantial amount of current.

<A1-1>

According to the comments, we performed additional experiments with carbon paper (CP) anodes. We confirmed that dCas9 cells produced only a modest level of photocurrent compared to slr0688i cells even when using a CP anode (Supplementary Fig. 8(a)). This result assures that the high photocurrent level of 0688i cells observed with a carbon paper anode (Fig. 1(e)) is attributed to the outer membrane deprivation but not to the property of carbon paper anodes, representing the same phenomenon as that observed with ITO anodes. It should be noted, however, that the photocurrent profile observed with a carbon paper anode may not be completely the same as that of an ITO anode, as we noticed that slr0688i cells generated photocurrent (ca. 2 to 3 $\mu\text{A}/\text{cm}^2$) even after replacing the culture supernatant to fresh BG11 medium (Supplementary Fig.8(b)); when using an ITO anode, this treatment completely abolished the photocurrent (Fig. 1(d); Supplementary Fig.8(b)). This implies that the photocurrent of slr0688i cells observed with carbon paper anodes might partially stem from electron transfer pathway(s) different from that in the case of ITO anodes. To elaborate on the above points, we added new Supplementary Fig. 8 on p. 8 of the revised SI and revised the text on lines 158-173 of the revised manuscript.

The description of the genetic engineered strain is still poor and not enough to gain the reader with the understanding of the differences.

<A1-2>

We now added new data to describe the phenotypes of slr0688i mutant in more detail as follows. I) Electron-microscopic images of both slr0688i and dCas9 cells are shown in Supplementary Fig. 1(a, b) on p. 1 of the revised SI, the former of which exhibits outer membrane detachment. II) Growth curves of both cells are also shown in Supplementary Fig. 1(c). The slr0688i cells shows slightly retarded growth compared to that of dCas9 cells. Also, we revised the manuscript to provide more information about the slr0688i mutant (lines 71-73, 83-85, 427, 433 and 751-756 of the revised manuscript).

Measuring NADPH using fluorescence as it was done here would be not sensitive enough to see the small differences generating the current. The enzymatic detection methods (sigma kit, for example) or the two-dimension fluorescent mapping method should be used.

<A1-3>

Our main purpose of employing NADPH fluorescence measurement was to reveal the action of photosynthetic inhibitors but not to connect the intracellular NADPH level to photocurrent generation, as we already confirmed that the outer membrane-deprivation does not significantly affect the intracellular NADP(H) content, which was quantified by the enzymatic detection method (Supplementary Fig. 14). To this aim, the effect of the inhibitors on time-dependent changes of the intracellular NADPH level before and after the light illumination gives critical information about their sites of action. Because the light-dependent NADPH formation and the following NADPH consumption under dark occur within a few seconds, we believe that the NADPH fluorescence monitoring has an advantage thanks to its good time-resolution over other methods such as enzymatic detection. In addition, we previously confirmed that NADPH fluorescence measurements could give the results qualitatively consistent with enzymatic NADP(H) detection method (Tanaka et al., 2021; reference 82).

Reviewer #2 (Remarks to the Author):

Dear Researchers!

Thank you very much for correcting most of my remarks. Now my doubts have been overwhelmingly dissolved. There are only some minor remarks I would like to perform.

Zhang et al. (1992) concluded that copper determines the ratio between plastocyanin and cytochrome c, however, 0.3 μM copper (the concentration of standard BG11 medium used in your paper) allows the expression of both carrier molecules, whereas no expression of cytochrome c was detected at 1 μM copper in the medium. Nevertheless Zhang et al. (1994, J Biol Chem 269, 5036-5042) stated that even 1 μM copper did not completely abolish the expression of cytochrome c, which explains that cytochrome c or plastocyanin single mutants can grow in conditions that strongly favor the expression of the deleted gene.

<A2-1>

We thank the reviewer's providing us the important information. Although plastocyanin is well known as a copper-dependent protein and the presence of 0.3 μM copper favors the expression of plastocyanin over cytochrome c, we now understood that high concentration of copper (e.g., 1 μM) cannot completely abolish the expression of cytochrome. In accordance with the reviewer's comment, we deleted the sentence on lines 1106-1108 of the revised manuscript.

Besides I would like to suggest to modify Fig. 2 to include the splitting of water below PSII as you did the reversion above COX and Cyd.

<A2-2>

In accordance with the reviewer's comment, we added the water-splitting reaction below PSII in Fig. 2(b) (on p. 67 of the revised manuscript).

You wrote that inside the yellow dashed line there are the components where EET is inhibited by GA but not by pCMB, however, the Calvin cycle is included within this line although it is marked to be inhibited by pCMB. I also wonder why you mark the oxidative pentose phosphate pathway (glycolysis and TCA cycle) in this figure to be inhibited by GA but not by pCMB although you think that pCMB also inhibits the oxidative pentose phosphate pathway (lines 418 – 421).

<A2-3>

We apologize for providing confusing marks. We revised Figure 2(b) and the manuscript (lines 1102-1103).

You write that the P700 oxidation level under illumination with far red light was elevated in GA-treated cells, however, this was even more true to PMA (supplementary figure 8). How does this fit with its proposed role to inhibit FNR. If no electrons can be transferred from Fd to NADP⁺ anymore, P700 should be in a more reduced state.

<A2-4>

We appreciate reviewer's providing us insightful comments on this issue. Ferredoxin (Fd) transfers electrons not only to ferredoxin-NADP⁺-oxidoreductase (FNR) but also to flavodiiron proteins (Setif et al., 2020, *Biochimica et Biophysica Acta - Bioenergetics*, 1861: 148256) or NDH-1 complex (Pan et al., 2020, *Nature Communications*, 11: 610; Schuller et al., 2019, *Science*, 363: 257-260; Zhang et al., 2020, *Nature Communications*, 11: 1-13); these routes could lead to oxidation of P700 under illumination even when FNR is inhibited. Furthermore, it is well known that the P700 oxidation level under FR light depends on the redox state of PQ governed by the respiration and/or cyclic electron transfer (CET) around PSI (Mi et al., 1992; reference 49). We now consider that the possibility that PMA acts pleiotropically on CET or respiration cannot be ruled out. Accordingly, we revised the text on lines 224-240 of the revised manuscript.

And why does PMA prevent the consumption of NADPH in the dark (supplementary figure 8) if it does not inhibit NDH1 or NDH2 (no such mark in fig. 2).

<A2-5>

As we described in A2-4, the possibility that PMA acts pleiotropically on CET or respiration cannot be ruled out; this may directly or indirectly prevent the consumption of NADPH under dark. As in A2-4, we revised the text on lines 224-240 of the revised manuscript.

Finally in line 156 of supplemental material you should add (green) after DCMU and split supplementary figure 8 into a, b and c as you refer to supplementary Fig. 8(a) in line 174 of your main manuscript. Besides it would be helpful to describe the oxidation state of the (OOH) in line 540 to allow the same total charge on both sides of the equation.

Overall I wish you once more good luck and hope you can publish it as soon as possible.

<A2-6>

According to the reviewer's comment, we've revised the following text:

Line 213 of the revised SI

Line 184 of the revised SI

Line 578 of the revised manuscript

Reviewer #3 (Remarks to the Author):

The revisions made by Nakanishi et al. have addressed most of the concerns of the referees. The authors are to be commended on these efforts, some of which are above and beyond. I think this manuscript is ultimately deserving to be published in Nat. Comm. if the main results can be validated; however there remain important points that remain poorly addressed by the authors regarding the electrochemical approach, which leaves this reviewer unconvinced by the interpretation of the data.

Specifically, these issues are:

1) The authors state: "We used cell suspensions placed by gravity onto the electrode for photocurrent measurement to focus exclusively on the EET activity of the cells but not on the cell-electrode interaction";

This reviewer is very concerned about the electrochemical approach and the interpretations of the results in Figure 1. If the authors want to "focus exclusively on the EET activity of the cells";, then the best practice would be to use a stirred cell suspension, perhaps with the addition of a non-cell permeable mediator to further aid the electron transfer from the solution bulk to the electrode. Only then would there be minimal cell-electrode interaction at play. However, in the current method (letting the cells settle on the electrodes over time with the aid of gravity), there will be inevitably a layer of cells interacting with the electrode with multi-layers above. It is the formation of this first layer of cells that contributes most to the photocurrent and the multilayers above will contribute minimally to the photocurrent. Therefore, very precise cell loadings to the device (as claimed by the authors) is not sufficient for providing information on how many cells and how the cell-electrode interface contribute to the photocurrent. This is problematic because the authors are making very precise quantitative conclusions (35-fold enhancement compared to wildtype etc), which is not appropriate using this qualitative approach for performing bioelectrochemistry. For the authors to attribute the photocurrent output differences to differences in the EET activity alone is therefore inappropriate (there is not enough information to support this).

Some suggestions for addressing this problem:

i) Unless the method can be revised (for example with properly immobilised cells), please rephrase the discussion of the results to be more transparent about the limitations of this cell-loading approach. I would suggest re-wording the title as well to remove the "35-fold enhancement",

ii) Take some representative SEM or microscopy images of the cells (for both wt and mutant) covering the bottom of the device to show information including how many cell multilayers are there? What is the total coverage of the electrode? It is homogenous distribution or not?

<A3-1>

Regarding the reviewer's comment on electrochemical approach, we additionally measured photocurrents from both stirred and non-stirred cell suspensions with the use of non-cell permeable mediator, ferricyanide (Supplementary Fig. 4). In both cases, where the effect of cell-electrode interaction on current detection was minimized, slr0688i generated significantly larger photocurrent than dCas9 (wild type). We added new Supplementary Fig. 4 on p. 4 of the revised SI and revised the text on lines 89-92, 115-119, 293-294 and 298 of the revised manuscript.

According to the reviewer's comment i), the title of the article was changed to "Order-of-magnitude enhancement in photocurrent generation of *Synechocystis* sp. PCC 6803 by outer membrane deprivation". Accordingly, we revised the text on lines 1, 25-26, 88-89, 119-120, 374-375, 380-381 of the revised manuscript.

Regarding the reviewer's comment ii), we added photographs of both wild type and slr0688i cells at the bottom of electrochemical chambers, as well as microscopy images of these cells on ITO electrodes (Supplementary Fig. 20). Cells appeared to be homogeneously distributed on the electrode surface, and the autofluorescence measurement along z-axis using a confocal laser scanning microscope suggested the majority of the cells reside within 50 to 70 μm distance from the electrode surface. Accordingly, we added new Supplementary Fig. 20 on p. 27-29 of the revised SI and revised the text on line 561-563 and 757-762 of the revised manuscript.

2) The interpretation of the chronoamperometry photocurrents is highly unconvincing. As mentioned by reviewer 3, typically, the fairest section for photocurrent comparison is at the steady-state region (after the initial peaks). The authors did not pick this steady state region for analysis (which according to Fig 1 (c) is at the 500s time point), instead picking the transient peak at around 70s time point.

The authors' response to this is that We observed that the photocurrent produced by the outer membrane-deprived cells did not necessarily show the initial peaks. However, it is clear looking at Fig 1 (c) and (e) that there is an initial peak- it is at 70 s, which is what the authors picked for analysis and cautioned against by reviewer 3.

<A3-2>

According to the comment, we analyzed 33 and 21 photocurrent traces of slr0688i and dCas9, respectively, including the steady state region and revised Fig. 1(b) on P. 62 of the revised manuscript. We confirmed ca. 20-fold enhancement of photocurrent at the steady state region. Also, as answered in A3-1, we modified the title of this manuscript to "Order-of-magnitude enhancement in photocurrent generation of *Synechocystis* sp. PCC 6803 by outer membrane deprivation". Accordingly, we revised the text on lines 1, 25-26, 88-89, 93-96, 119-120, 374-375, 380-381 of the revised manuscript.

Even more problematic is the interpretation of the new carbon cloth data Fig 1 (e). It can be seen that after 'light on', the dark current does not return to baseline, but instead stays very high - very similar to the photocurrent. Unless the authors can show the dark current returning fully to the original baseline, they cannot claim that the 30 μAcm^{-2} photocurrent output that they put forward as the new benchmark output.

Overall, although the non-electrochemical data put forward in this manuscript convincingly shows that the mutant cell line has higher EET activity, the electrochemical data put forward is not sufficiently convincing to back the quantitative claims of the paper and instill confidence regarding the reproducibility of the outputs.

<A3-3>

Regarding Reviewer 3's comment on experiments with a carbon paper anode, we performed additional experiments and confirmed that the dark current returned fully to the original level (Fig. 1(e); Supplementary Fig. 7). When using a carbon paper anode, the current level tended to remain high after switching off the light (Fig. 1(e)), and it took approx. 1 h to decline back to the basal level. We speculate that this is probably because pores of carbon paper (about 100 μm) prevent mediators reduced by cells from diffusive dispersion into the bulk electrolyte and/or trap cells so densely that sufficient CO_2 is not available, leading to the overaccumulation of intracellular reducing equivalents which could possibly reduce mediators even in the dark. We replaced the original figure with the newly obtained data (Fig. 1(e) on p. 62 of the revised manuscript), added new Supplementary Fig. 7 on p. 7 of the revised SI, and revised the text on line 156 of the revised manuscript.

Reviewers' Comments:

Reviewer #1:

Remarks to the Author:

3ed cycle review

The 3ed version of the ms is improved and much work has been done to answer the inquiries. As expected, the use of carbon paper for the interacting anode resulted with a fundamental and significant increase in the amount of produced current. Since the title is "tenfold increase in the current" It is not clear why not simply repeat the experiments with this anode and making the results much more significant and trustable while getting 20-40 microAmp.cm⁻² currents. I recall that a comment from the first round was not related in the revisions: "It was already shown that manipulating the cell membrane by physical means like an osmotic treatment or low-pressure microfluidizer treatment significantly enhances the EET in *Synechocystis* (ref 15. Saper et al Nat Comm). Therefore, this study repeats this phenomenon that was previously obtained by physical means by the application of a genetic eng manipulation technique". This should be described in this manuscript since most probably the physical treatments induced similar phenotype of the membrane deprivation, obtained here by genetic manipulation. In ref 15 the current was increased from 10 to 30 microAmp.cm⁻² . Here, in this work, it was going from 0 to about 0.2 microAmp.cm⁻² with ITO or to 30 with CP. Therefore, the sentences in this ms telling that "this is the highest current obtained with genetic eng cyanobacteria" is related to the gen eng cyano and not to the amount of current. Since the photocurrent is induced by light, either PSII or PSI or both are involved. In this case DCNU inhibits the current and therefore PSII is involved and reduces the PQ pool that is shared by the respiratory and photosynthetic electron transfer. NADPH is produced and probably is the mediator. It is not expected that with the low currents obtained here with the ITO system changes in the amount of NADPH could be detected in the methods used here. The two dimension fluorescence mapping and the carbon anode system should be used (ref 17). Lane 63 "Here, we hypothesize an additional reason for the extremely low EET activity of cyanobacteria: the low permeability of the outer membrane." This is the mechanism, not additional reason for

Fig 1. Missing legend to fig 1e. What was the result with dCas. Why there is no plateau reached?

The figures are not easy to follow. Why not indicating in the panels the inhibitors, strains etc. It would be much easier to observe.

What is the difference between sup Fig 8a and sup fig 7 and Fig 1e? Why there is only 3 microA/cm² in Fig 8 as compared to about 30 in the others?

Line 1104. The yellow square does not surround PSII, the target of DCMU as it is says in the legend.

Reviewer #2:

Remarks to the Author:

Dear researchers,

thank you very much for having corrected most of my remarks. Thank you also for adding the growth curves in supplementary figure 1, however, I wonder did you really reach an OD730 over 10.0. Normally cells start to shadow themselves if a certain density (e.g. OD730 = 5.0) is reached thereby preventing further growth.

In the legend of figure 1b you write about black and red circles. I do not see any circles but rather

points, which are vertically elongated with the bars of errors and connected to each other to form a continuous line. Is the radius of the circles that small that they appear as points. Otherwise you should correct the terminus "circles". You state that slr0688i generated 40 and 20 times as much photocurrent, however, since the measured values of dCas are that low it may be advantageous to add a table of the values of both slr0688i and dCas9 at certain intervals of time.

In fig. 2b I still do not completely understand the yellow dashed line. You state that inside the line the components of light dependent EET are inhibited by DCMU, KCN, PMA and GA but not by pCMB, however, no component completely inside the line is marked to be inhibited by any of them. The oxidative pentose phosphate pathway, which is drawn partly within, is inhibited by pCMB. The light dependent electron transport chain from photosystem II to Fd and FNR is completely outside of the yellow square since the most important components within (NDH1, NDH2 and SDH) are more involved in (dark) respiration.

For some experiments in the material and method section you state that negative controls are performed with the same volume of the corresponding solvent (here DMSO for pCMB). I think you should add this information for all experiments if this is the case. In line 602 when writing about the different inhibitors you should explain the solvent(s) and the concentrations of the corresponding stock solutions of all of them. Are the negative controls in fig. 2 or supplementary fig. 10 performed with the same cells before the actual experiment after adding the inhibitor was performed and therefore every single experiment has its distinct negative control? Otherwise the/some experiments can be summarized in one graphic if no solvent or the same has been used for the/some experiments. Do the white (empty) circles in Supplementary figure Fig 10 with error bars represent the negative control without the inhibitor? It is stated nowhere.

With the legend of supplementary figure No 12 I am rather confused. What are the left and right panels? Is this description referring to parts of the figure that have already been deleted from your manuscript? Also the description of the time scale should be uniform, either ms or msec at all three subfigures. Besides it might be helpful to add (a), (b) and (c) in lines 188 and 189.

You answered me that Fd can transfer electrons to NDH1. Does NDH transfer electrons from Fd to NAD(P) thereby creating NAD(P)H as drawn in figure 2 or can NDH1 directly transfer electrons from Fd to NDH1 without an NAD(P)H intermediate? If there is an NAD(P)H intermediate it would contradict your statement in line 463 that the corresponding metabolic pathways are not linked to NADP reduction, since in that case a mutant of NDH complex would exhibit an enhanced EET linked to a more reduced NAD(P)H pool. Besides you stated that a mutant lacking a functional NDH complex exhibits an enhanced EET (line 467), however, this fact was not discovered in the cited paper (Bradley et al., 2013). In this paper they cited Cournac et al. (2004) to have found out that this mutant has a smaller but highly reduced NAD(P)H pool and improved H₂ production. Neither Cournac et al. (2004) wrote explicitly about enhanced EET of this mutant.

Overall I wish you good luck for publishing your wonderful paper.

Reviewer #3:

Remarks to the Author:

The authors have addressed all concerns of the reviewers, the article can now be accepted for publication.

In the manuscript, we have highlighted the revised text in pale green (to be distinguished from those already marked in yellow and pale blue in NCOMMS-21-01109B and NCOMMS-21-01109C, respectively). Text deleted from the older versions of the manuscript is highlighted in gray. Below we provide our point-by-point responses to each of the reviewers' comments.

Reviewer #1 (Remarks to the Author):

3ed cycle review

The 3ed version of the ms is improved and much work has been done to answer the inquiries.

As expected, the use of carbon paper for the interacting anode resulted with a fundamental and significant increase in the amount of produced current. Since the title is "tenfold increase in the current"; It is not clear why not simply repeat the experiments with this anode and making the results much more significant and trustable while getting 20-40 microAmp.cm⁻² currents.

<A1-1>

According to the comment, we repeated the experiments, i.e., examination of the effects of photosynthesis inhibitors, using carbon paper anodes. We confirmed that the effects of inhibitors were almost the same as those observed with ITO anodes, reinforcing the conclusion obtained from ITO-based experiments. Accordingly, we added a new Supplementary Fig. 15 to the revised SI and revised the text on lines 265-270 of the revised manuscript accordingly. Please note that, as we also mentioned in the revised manuscript (lines 164-173) and this point-by-point response A1-8, we used dilute cell suspension of mid-exponential growth phase for this type of comparative analysis, i.e., investigation of the photosynthesis inhibitors (Supplementary Fig. 15) or the comparison of photocurrent between slr0688i and dCas9 cells (Supplementary Fig. 8(a)).

I recall that a comment from the first round was not related in the revisions: “It was already shown that manipulating the cell membrane by physically means like an osmotic treatment or low-pressure microfluidizer treatment significantly enhances the EET in *Synechocystis* (ref 15. Saper et al Nat Comm). Therefore, this study repeats this phenomenon that was previously obtained by physical means by the application of a genetic eng manipulation technique”; This should be described in this manuscript since most probably the physical treatments induced similar phenotype of the membrane deprivation, obtained here by genetic manipulation.

<A1-2>

According to the comment, we described and discussed the differences between the “physically-treated” cells and the outer membrane-deprived cells in the revised manuscript (lines 457 to 493). We understand that a gentle physical treatment, which presumably causes a modest damage(s) to the cell membranes, leads to a higher electrogenic activity and thus the study by Saper et al. (2018) (reference 15) provides a novel approach to improve the bio-photoelectrochemical system. However, although both the “physically-treated” cells and the outer membrane-deprived cells presented by our study show the enhanced electrogenic activity, the underlying mechanisms are considered different from each other for the following reasons.

(1) The gentle physical treatments by Saper et al. i.e., passing through a low-pressure microfluidizer at 10-15 psi under the presence of 400 mM NaCl or providing a simple osmotic shock to the cells using 400 mM NaCl, might partially perturb the structural integrity of the cell surface but is not expected to deprive the outer membrane from the cells because the cyanobacterial outer membrane is well known to be tightly anchored to the peptidoglycan even after the mechanical disruption of the cells, such as passing through a French pressure cell press at 20,000 psi or mechanical cracking (Jurgens et al., 1983; reference 61; Weckesser and Jurgens, 1988; reference 62; Kowata et al., 2017; reference 20). Detaching the outer membrane thus requires the enzymatic degradation of the peptidoglycan and/or solubilization of the outer membrane by detergent treatment (e.g., Jurgens and Weckesser, 1985: reference 63). Please note that the outermost surface of *Synechocystis* cells is the S-layer-like proteinaceous layer consisting of Sll1951 protein (Trautner and Vermaas 2013, J Bacteriol 195:5370); therefore, the SEM images shown by Saper et al. do not provide information about the structural status of outer membrane.

(2) Suspending the cells in 400 mM NaCl unequivocally generates the osmotic pressure to the cytoplasmic membrane rather than to the outer membrane because the outer membrane contains abundant non-specific channels those allow the rapid passage of ions across the outer membrane (Kowata et al., 2017). Indeed, Reed et al (reference 64) showed that treating the cells with 490 mM NaCl affected the cytoplasmic membrane and resulted in the leakage of various cytosolic metabolites, which may include the possible mediator(s) for EET.

(3) Genetically engineered outer membrane deprivation avoids interfering the cytoplasmic membrane

function and cytosolic metabolites release, so that even abundant cytosolic molecules, such as NADP(H), was not detected in the culture supernatant of the outer membrane-deprived cells (Kojima and Okumura, 2020; reference 25). This was experimentally reassured in the present study (Supplementary Fig. 21), which will be discussed in detail below in A1-4.

From these facts, we do not perceive the indication that the effect of a gentle physical treatment can be reproduced by the genetically engineered outer membrane deprivation. Rather, it appears the most reasonable to assume that the electron transfer path from the physically-treated cells and that from the outer membrane-deprived cells, to the electrodes, are different; as a matter of fact, this assumption was verified by a striking difference in their photocurrent properties that DCMU enhances the photocurrent of the former cells but abolishes the photocurrent from the latter cells. If our present study replicated the “physical treatment”, the DCMU-enhanced photocurrent should have been observed in our experiments.

Furthermore, following facts seems noteworthy. The effect of physical treatment is assumed not to be long-lasting but temporal, because living cells are intrinsically self-repairing (e.g., the case of membrane-damaged *Escherichia coli*; Chilton et al., 2001, *International Journal of Food Microbiology*, 71: 101–104). In contrast, genetically manipulated cells can proliferate while the effect of outer membrane deprivation is maintained. This difference is potentially decisive when considering the robustness of biophotovoltaics, especially from the point of view of biotechnological applications.

In ref 15 the current was increased from 10 to 30 $\mu\text{A}\cdot\text{cm}^{-2}$. Here, in this work, it was going from 0 to about 0.2 $\mu\text{A}\cdot\text{cm}^{-2}$ with ITO or to 30 with CP. Therefore, the sentences in this ms telling that “this is the highest current obtained with genetic eng cyanobacteria”; is related to the gen eng cyano and not to the amount of current.

<A1-3>

According to the above comment, we revised the text on lines 159-164 of the revised manuscript.

Since the photocurrent is induced by light, either PSII or PSI or both are involved. In this case DCNU inhibits the current and therefore PSII is involved and reduces the PQ pool that is shared by the respiratory and photosynthetic electron transfer. NADPH is produced and probably is the mediator. It is not expected that with the low currents obtained here with the ITO system changes in the amount of NADPH could be detected in the methods used here. The two dimension fluorescence mapping and the carbon anode system should be used (ref 17).

<A1-4>

We appreciate Reviewer 1's insightful suggestion. We analyzed supernatants of slr0688i cells before and after the electrochemical measurement with a CP anode by means of the two-dimensional fluorescence mapping (2D-FM), enzymatic analysis, and LC-MS/MS analysis (Supplementary Fig. 21). Unfortunately, we failed to detect NADP(H). The 2D-FL revealed the presence of strong fluorescent molecules and one of which exhibited the fluorescence pattern of excitation wavelength (Ex) of 360 nm and emission wavelength (Em) of 450 nm, which was similar to that of NADPH (Ex 340 nm/Em 460 nm). However, the small discrepancy of Ex/Em led us to further analyze whether this fluorescence was actually derived from NADPH. As a matter of fact, neither an enzymatic assay (detection limit = 0.0625 μ M) nor a LC-MS/MS assay using the 50-fold concentrated supernatant detected NADPH. On the basis of these results, we concluded that the fluorescent molecules found in the supernatant was not NADPH. The absence of NADPH in the supernatant is reasonable because the bacterial periplasm is well known to be an oxidizing environment, and bacteria possess various systems to convey reducing power to periplasmic space rather than directly exporting the cytosolic reducing equivalents such as NAD(P)H (one of the examples can be found in Depuydt et al. (2009); reference 68). We added new Supplementary Fig. 21 to the revised SI and revised the text on lines 535-554, 794-797 and 847-869 of the revised manuscript.

Lane 63 “Here, we hypothesize an additional reason for the extremely low EET activity of cyanobacteria: the low permeability of the outer membrane.“; This is the mechanism, not additional reason for …..

<A1-5>

According to the comment, we replaced "reason" with "mechanism" in the revised manuscript (lines 60 and 64).

Fig 1. Missing legend to fig 1e. What was the result with dCas. Why there is no plateau reached?

<A1-6>

We apologize for our mistake that in the previous manuscript Fig. 1(e) was described as Fig. 1(d) in the legend. We corrected it on line 1191 in the revised manuscript. Regarding the data, the photocurrent actually reached a plateau after ca. 750 sec of the measurement. To make this clearer, we added the inset showing the photocurrent between 0 to 1,200 sec of the measurement and revised the figure legend (line 1194). Please note that during the previous review cycle we added the figures (Supplementary Fig. 8) for detailed comparison of photocurrent with CP anodes between slr0688i and dCas9 cells, and the result was described in the manuscript (line 173-177).

The figures are not easy to follow. Why not indicating in the panels the inhibitors, strains etc. It would be much easier to observe.

<A1-7>

According to the comment, we modified the following figures:

Fig. 1(b-d), 2(a), 3, 4; Supplementary Fig1(b), 2, 4, 6, 8, 9, 10, 11, 12, 13, 14, 15, 17, 18, 19, 20

What is the difference between sup Fig 8a and sup fig 7 and Fig 1e? Why there is only 3 microA/cm² in Fig 8 as compared to about 30 in the others?

<A1-8> The purpose of Fig. 1(e) and Supplementary Fig. 7 was to demonstrate the absolute value of the photocurrent can be improved to the level highest among the BPV systems using the untreated whole cells of *Synechocystis*. To this aim, we utilized the dense cell suspension of slr0688i, i.e., 6 days-cultivated cell culture; this successfully achieved the high photocurrent density. However, at the same time we noticed that this dense and “aged” cell culture sometimes produced unstable photocurrent for unknown reason. On the other hand, we found that the dilute cell suspension of mid-exponential growth phase, i.e., 3 days-cultivated cell culture, tends to show more stable and reproducible photocurrent although the absolute photocurrent value was lower than that of dense, aged cells. As such, for comparative analysis of photocurrent, i.e., the comparison of photocurrent between slr0688i and dCas9 cells and investigation of the photosynthesis inhibitors (Supplementary Fig. 8(a) and 15, respectively), we employed this dilute and “young” cells. To make this clearer and to avoid providing misleading information with readers, we revised the text on lines 164-173 of the revised manuscript.

Line 1104. The yellow square does not surround PSII, the target of DCMU as it says in the legend.

<A1-9> We'd like to apologize for the confusing schematic and sentences in the legend. We modified Fig. 2(b), and now the yellow dashed square indicates the cellular components or metabolism which apparently keep providing reducing equivalents under illumination in the presence of pCMB but not in the presence of other inhibitors (DCMU, KCN, PMA and GA). Accordingly, we revised the text on lines 257-258 and 1230-1233 of the revised manuscript.

Reviewer #2 (Remarks to the Author):

Dear researchers,

thank you very much for having corrected most of my remarks. Thank you also for adding the growth curves in supplementary figure 1, however, I wonder did you really reach an OD₇₃₀ over 10.0. Normally cells start to shadow themselves if a certain density (e.g. OD₇₃₀ = 5.0) is reached thereby preventing further growth.

<A2-1>

We consider that the culture volume, the shape and size of culture vessels, and shaking speed affect cyanobacterial growth. In our experiments, we usually used 100 to 125 ml volumetric flasks containing 12 mL of culture, shaken at 140 rpm. This condition achieves a high surface area of culture medium, which avoids self-shadowing and maintains good aeration; OD₇₃₀ of the cell cultures generally exceeds 10 within a week. If culture medium were contained in smaller flasks or test tubes, OD₇₃₀ would not exceed 10 as the reviewer pointed out above. Accordingly, we briefly modified the text in the Materials and Methods (lines 615-616).

In the legend of figure 1b you write about black and red circles. I do not see any circles but rather points, which are vertically elongated with the bars of errors and connected to each other to form a continuous line. Is the radius of the circles that small that they appear as points. Otherwise you should correct the terminus “circles”.

<A2-2>

We replaced “circles” with “points” on line 1181 in the revised manuscript.

You state that slr0688i generated 40 and 20 times as much photocurrent, however, since the measured values of dCas are that low it may be advantageous to add a table of the values of both slr0688i and dCas9 at certain intervals of time.

<A2-3>

According to the comment, we added Supplementary Table. 1 to show the averaged current density values of both slr0688i and dCas9 cells and revised the text on line 96 of the revised manuscript.

In fig. 2b I still do not completely understand the yellow dashed line. You state that inside the line the components of light dependent EET are inhibited by DCMU, KCN, PMA and GA but not by pCMB, however, no component completely inside the line is marked to be inhibited by any of them. The oxidative pentose phosphate pathway, which is drawn partly within, is inhibited by pCMB. The light dependent electron transport chain from photosystem II to Fd and FNR is completely outside of the yellow square since the most important components within (NDH1, NDH2 and SDH) are more involved in (dark) respiration.

<A2-4>

We'd like to apologize for the confusing schematic and sentences in the legend. As answered in A1-9, We modified Fig. 2(b), and now the yellow dashed square indicates the cellular components or metabolism which apparently keep providing reducing equivalents under illumination in the presence of pCMB but not in the presence of other inhibitors (DCMU, KCN, PMA and GA). Accordingly, we revised the text on lines 257-258 and 1230-1233 of the revised manuscript.

For some experiments in the material and method section you state that negative controls are performed with the same volume of the corresponding solvent (here DMSO for pCMB). I think you should add this information for all experiments if this is the case. In line 602 when writing about the different inhibitors you should explain the solvent(s) and the concentrations of the corresponding stock solutions of all of them.

<A2-5>

According to the comment, we added the explanations of the control experiments (lines 694-696 of the revised manuscript and lines 118-119, 142-144 and 211-213 of the revised SI). Moreover, we provided information on preparation of inhibitor solutions in the revised manuscript (lines 621-627).

Are the negative controls in fig. 2 or supplementary fig. 10 performed with the same cells before the actual experiment after adding the inhibitor was performed and therefore every single experiment has its distinct negative control? Otherwise the/some experiments can be summarized in one graphic if no solvent or the same has been used for the/some experiments.

<A2-6>

As the reviewer pointed out above, in Fig. 2 and Supplementary Fig. 10, every single experiment has its distinct negative control.

Do the white (empty) circles In Supplementary figure Fig 10 with error bars represent the negative control without the inhibitor? It is stated nowhere.

<A2-7>

According to the comment, we added to the revised manuscript the explanation that white circles indicate respective controls (lines 186-187).

With the legend of supplementary figure No 12 I am rather confused. What are the left and right panels? Is this description referring to parts of the figure that have already been deleted from your manuscript? Also the description of the time scale should be uniform, either ms or msec at all three subfigures. Besides it might be helpful to add (a), (b) and (c) in lines 188 and 189.

<A2-8>

We apologize for the confusing description. According to the comment, we revised Supplementary Fig. 12 and its figure legend (lines 184-185 and 188-196 of the revised SI).

You answered me that Fd can transfer electrons to NDH1. Does NDH transfer electrons from Fd to NAD(P) thereby creating NAD(P)H as drawn in figure 2 or can NDH1 directly transfer electrons from Fd to NDH1 without an NAD(P)H intermediate? If there is an NAD(P)H intermediate it would contradict your statement in line 463 that the corresponding metabolic pathways are not linked to NADP reduction, since in that case a mutant of NDH complex would exhibit an enhanced EET linked to a more reduced NAD(P)H pool.

<A2-9>

Following literatures show that electron transfer from Fd to NDH-1 complex does not yield NAD(P)H intermediate: Pan et al., 2020, Nature Communications, 11: 610; Schuller et al., 2019, Science, 363: 257-260; Zhang et al., 2020, Nature Communications, 11: 1-13.

Besides you stated that a mutant lacking a functional NDH complex exhibits an enhanced EET (line 467), however, this fact was not discovered in the cited paper (Bradley et al., 2013). In this paper they cited Cournac et al. (2004) to have found out that this mutant has a smaller but highly reduced NAD(P)H pool and improved H₂ production. Neither Cournac et al. (2004) wrote explicitly about enhanced EET of this mutant.

<A2-10>

In reference 12 (Bradley et al., 2013), extracellular ferricyanide reduction rates for M55 mutant lacking NDH-1 complex is reported to be “dramatically higher than wild-type rates in dark and light conditions”. Therefore, according to the comment, we replaced “EET” with “ferricyanide reduction” in line 528-529 in the revised manuscript.

Overall I wish you good luck for publishing your wonderful paper.

Reviewer #3 (Remarks to the Author):

The authors have addressed all concerns of the reviewers, the article can now be accepted for publication.

Reviewers' Comments:

Reviewer #1:

Remarks to the Author:

The authors have done a great job reviewing the manuscript and making the requested experiments, as well as answering the queries.

Reviewer #2:

Remarks to the Author:

Dear researchers,

thank you very much for having addressed and corrected all of my remarks now. I think this paper has to be published as soon as possible since there is a great economic output to improve biophotovoltaics. There is only one minor thing left I want to tell you. It was a great discovery that pCMB could increase photocurrent in contrast to the other tested inhibitors. Therefore you want to explain why pCMB differs from the others especially from GA, since both have been reported to inhibit the Calvin cycle. From your second draft on you started with this yellow dashed line in Fig. 2 to summarize the components that are inhibited by GA (in one version only by this molecule, in another version also by DCMU, PMA and KCN) but not by pCMB but every time I had some critical remarks on this yellow dashed line. I know you want to explain the distinct role of pCMB and you are very proud of your invention of this line to separate some components of the electron transport chain from others according to their inhibition by pCMB, however, the explanation of this yellow dashed line never really fitted to the figure and its marked sites, where the inhibitors interact with the components of the electron transport chain. Even in your last version you do not include the components of the thylakoid membrane from photosystem II to photosystem I, which are all inhibited by various molecules but not by pCMB according to this figure.

If you insist on this dashed line I will accept it because you did a lot of work and improved figure 2 compared to previous versions. The explanation of the line is now right because the surrounded components are not inhibited by pCMB but by others (although only by one e.g. glycolysis by GA) according to the marked sites of the inhibitors' effects. However, I think the increased photocurrent of the mutant strain of *Synechocystis* sp. PCC 6803 lacking its outer membrane and its further enhancement by pCMB is a great discovery for its own and you need not explain the molecular mechanism underlying the special role effect of pCMB in all details, if you do not know (yet) exactly. This does not minimize the success of your work in any way but on the contrary may stimulate you or other researchers in the future to reveal these mechanisms. So I will not mind whether you keep this dashed line or not. Anyway I think you can publish it now.

I wish you good luck

Best regards

➤ Below we provide our point-by-point responses to the reviewers' comments.

Reviewer #1 (Remarks to the Author):

The authors have done a great job reviewing the manuscript and making the requested experiments, as well as answering the queries.

Reviewer #2 (Remarks to the Author):

Dear researchers,

thank you very much for having addressed and corrected all of my remarks now. I think this paper has to be published as soon as possible since there is a great economic output to improve biophotovoltaics. There is only one minor thing left I want to tell you. It was a great discovery that pCMB could increase photocurrent in contrast to the other tested inhibitors. Therefore you want to explain why pCMB differs from the others especially from GA, since both have been reported to inhibit the Calvin cycle. From your second draft on you started with this yellow dashed line in Fig. 2 to summarize the components that are inhibited by GA (in one version only by this molecule, in another version also by DCMU, PMA and KCN) but not by pCMB but every time I had some critical remarks on this yellow dashed line. I know you want to explain the distinct role of pCMB and you are very proud of your invention of this line to separate some components of the electron transport chain from others according to their inhibition by pCMB, however, the explanation of this yellow dashed line never really fitted to the figure and its marked sites, where the inhibitors interact with the components of the electron transport chain. Even in your last version you do not include the components of the thylakoid membrane from photosystem II to photosystem I, which are all inhibited by various molecules but not by pCMB according to this figure. If you insist on this dashed line I will accept it because you did a lot of work and improved figure 2 compared to previous versions. The explanation of the line is now right because the surrounded components are not inhibited by pCMB but by others (although only by one e.g. glycolysis by GA) according to the marked sites of the inhibitors' effects. However, I think the increased photocurrent of the mutant strain of *Synechocystis* sp. PCC 6803 lacking its outer membrane and its further enhancement by pCMB is a great discovery for its own and you need not explain the molecular mechanism underlying the special role effect of pCMB in all details, if you do not know (yet) exactly. This does not minimize the success of your work in any way but on the contrary may stimulate you or other researchers in the future to reveal these mechanisms. So I will not mind whether you keep this dashed line or not. Anyway I think you can publish it now.

I wish you good luck

Best regards

- We greatly appreciate the reviewer's helpful comment. As the reviewer pointed out, we have focused on the intriguing action of pCMB and we thought it would be helpful to readers to illustrate the difference between pCMB and the other inhibitors. We believe that our illustration, i.e., the yellow dashed square, was not literally wrong, but according to the reviewer's comment we now decided to delete it to avoid conveying complicated information which may reduce the readability.